# CEBaB: Estimating the Causal Effects of Real-World Concepts on NLP Model Behavior

**Eldar David Abraham**[*,1]
eldar.a@campus.technion.ac.il

**Karel D'Oosterlink**[*,2,3]
Karel.DOosterlinck@UGent.be

**Amir Feder**[*,1]
feder@campus.technion.ac.il

**Yair Gat**[*,1]
yairgat@campus.technion.ac.il

**Atticus Geiger**[*,2]
atticusg@stanford.edu

**Christopher Potts**[*,2]
cgpotts@stanford.edu

**Roi Reichart**[*,1]
roiri@technion.ac.il

**Zhengxuan Wu**[*,2]
wuzhengx@stanford.edu

[1]**Technion – Israel Institute of Technology**    [2]**Stanford University**    [3]**Ghent University**

## Abstract

The increasing size and complexity of modern ML systems has improved their predictive capabilities but made their behavior harder to explain. Many techniques for model explanation have been developed in response, but we lack clear criteria for assessing these techniques. In this paper, we cast model explanation as the causal inference problem of estimating causal effects of real-world concepts on the output behavior of ML models given actual input data. We introduce CEBaB, a new benchmark dataset for assessing concept-based explanation methods in Natural Language Processing (NLP). CEBaB consists of short restaurant reviews with human-generated counterfactual reviews in which an aspect (food, noise, ambiance, service) of the dining experience was modified. Original and counterfactual reviews are annotated with multiply-validated sentiment ratings at the aspect-level and review-level. The rich structure of CEBaB allows us to go beyond input features to study the effects of abstract, real-world concepts on model behavior. We use CEBaB to compare the quality of a range of concept-based explanation methods covering different assumptions and conceptions of the problem, and we seek to establish natural metrics for comparative assessments of these methods.

## 1 Introduction

Explaining model behavior has emerged as a central goal within ML. In NLP, models have grown in size and complexity, and while they have become increasingly successful, they have also become more opaque [28,36], raising concerns about trust [18,23], safety [1,34], and fairness [16,19]. These concerns will persist if these models remain "black-boxes".

Seeking to open the black-box, researchers have developed methods that try to explain model behavior [2,11,13,30,41]. However, there is no consensus about how to evaluate such methods to allow robust

---

[*]Equal contribution. Author names alphabetical.

36th Conference on Neural Information Processing Systems (NeurIPS 2022).

Table 1: Toy examples illustrating the structure of CEBaB (actual corpus examples are longer and more complex; a sample is given in Appendix B). Beginning from an OpenTable review, we give crowdworkers an actual restaurant review and they generate counterfactual restaurant reviews that would have been written if some aspect of the dining experience were changed and all else were held constant. Five different crowdworkers labeled each of the actual and counterfactual texts according to their aspect-level sentiment and overall sentiment. Aspect level sentiment labels are three way: '+' (positive sentiment), '–' (negative), and 'unk' (the aspect's value is not expressed in the text). Overall sentiment labels are 1 (worst) to 5 (best). Edited aspect labels are shown in blue.

| | | food | ambiance | service | noise | overall |
|---|---|---|---|---|---|---|
| **Original text:** | Excellent lobster and decor, but rude waiter. | + | + | – | unk | 4 |
| **Edit Goal** | | | | | | |
| food: – | Terrible lobster, excellent decor, but rude waiter. | – | + | – | unk | 2 |
| food: unk | Excellent decor, but rude waiter. | unk | + | – | unk | 3 |
| ambiance: – | Excellent lobster, but lousy decor and rude waiter. | + | – | – | unk | 3 |
| ambiance: unk | Excellent lobster, but rude waiter. | + | unk | – | unk | 3 |
| service: + | Excellent lobster and decor, and friendly waiter. | + | + | + | unk | 5 |
| service: unk | Excellent lobster and decor. | + | + | unk | unk | 5 |
| noise: + | Excellent lobster, decor, and music, but rude waiter. | + | + | – | + | 4 |
| noise: – | Excellent lobster and decor, but rude waiter, and noisy. | + | + | – | – | 3 |

comparisons. This is not surprising, since such evaluations require very rich empirical data. Intuitively, we would like to (1) intervene on model inputs, to modify specific concepts without changing other correlated information, (2) observe the effects this has on model predictions, and, finally, (3) assess explanation methods for their ability to accurately predict these effects.

The absence of interventional data, or even an agreed-upon non-interventional benchmark, has created an environment in which explanation methods are often evaluated individually, and without comparison to alternatives. Attempts have been made to conduct comparative evaluations [11,17,37], but only with synthetic, simplified datasets. Furthermore, these attempts do not define a unified evaluation approach, nor do they seek to contribute benchmark datasets that support such evaluations.

In this paper, we seek to overcome this obstacle by introducing **CEBaB** (**C**ausal **E**stimation-**Ba**sed **B**enchmark). Table 1 summarizes the structure of CEBaB with a toy example: beginning with a review text from the OpenTable website, we crowdsourced edits of the original text that are designed to meet a specific goal, such as changing the food rating in the original text to negative or unknown. All of the resulting edits were validated by five crowdworkers and each full text was evaluated by five crowdworkers for its overall sentiment. CEBaB is grounded in 2,299 original reviews, which were expanded via this editing procedure to a total of 15,089 texts, targeting four different aspect-level concepts (food, service, ambiance, noise) with three potential labels (positive, negative, and unknown, i.e., not expressed in the review), and each full text was labeled on a five-star scale.

We focus on using CEBaB to compare concept-based explanation methods. This allows us to go beyond the effect of individual tokens to study how more abstract concepts (in our case, aspect-level sentiment) contribute to model predictions (about the overall sentiment of the text). Our proposed metrics center around assessing concept-based explanation methods for their ability to accurately estimate *causal concept effects* [17], allowing us to isolate the effect of individual concepts.

More specifically, we use CEBaB to measure the causal effects of particular variables in a causal graph, and we cast each explanation method as a causal estimator of these measurements. For example, suppose our causal graph of the data says that all four of our aspect-level categories will affect a reviewer's overall rating. To estimate the effect of positive food quality on the predicted overall rating from a classifier, we need to compare examples with high food quality to those with low quality, holding all other aspects constant. Such pairs of examples are normally not observed, but this is precisely what CEBaB provides. With CEBaB, we can directly compare the actual change in model predictions with the change that a concept-based explanation method predicts.

In our experiments, we evaluate five leading concept-based explanation methods: CONEXP [17], TCAV [26], ConceptSHAP [57], INLP [40], CausaLM [11], and S-Learner [27]. These methods make a wide range of different assumptions about how much access we have to the model's internal structure, and they also diverge in the degree to which they account for the causal nature of the

concept effect estimation problem. Remarkably, CEBaB reveals that most methods cannot beat a simple baseline. Indeed, this negative result emphasizes the value in our primary contribution of providing the data and metrics that enables a direct comparison of explanation methods.

## 2 Previous Work

**Benchmarks for Explanation Methods**  Benchmark datasets have propelled ML forward by creating shared metrics that predictive models can be evaluated on [22,25,54,55]. Unfortunately, benchmarks that are suitable for assessing the quality of model explanations are still uncommon [10, 21]. Previous work on comparing explanation methods has generally only correlated the performance of a given explainability method with others, without ground-truth comparisons [8,20,21,43].

Other works that do compare to some ground-truth either employ a non-causal evaluation scheme [26], use causal evaluation metrics which do not capture performance on individual examples [52], evaluate on synthetic counterfactuals and rule-based augmentations [11,52], or are tailored for a specific explanation method and hard to generalize [57]. To the best of our knowledge, CEBaB is the first large-scale naturalistic causal benchmark with interventional data for NLP.

**Explanation Methods and Causality**  Probing is a relatively new technique for understanding what model internal representations encode. In probing, a small supervised [6,51] or unsupervised [5,32,44] model is used to estimate whether specific concepts are encoded at specific places in a network. While probes have helped illuminate what models (especially pretrained ones) have learned from data, Geiger et al. [15] show with simple analytic examples that probes cannot reliably provide causal explanations for model behavior.

Feature importance methods can also be seen as explanation methods [33]. Many methods in this space are restricted to input features, but gradient-based methods can often quantify the relative importance of hidden states as well [3,46,48,58]. The Integrated Gradients method of Sundararajan et al. [50] has a natural causal interpretation stemming from its exploration of baseline (counterfactual) inputs [15]. However, even where these methods can focus on internal states, it remains difficult to connect their analyses with real-world concepts that do not reduce to simple properties of inputs.

Intervention-based methods involve modifying inputs or internal representations and studying the effects that this has on model behavior [30, 41]. Recent methods perturb input or hidden representations to create counterfactual states that can then be used to estimate causal effects [9,12,47,53,15]. However, these methods are prone to generating implausible inputs or network states unless the interventions are carefully controlled [14].

Generating counterfactual texts automatically remains challenging and is still a work-in-progress [4]. To overcome this problem, another class of approaches proposes to manipulate the representation of the text with respect to some concept, rather than the text itself [9,11,40]. These methods fall into the category of concept-based explanations and we discuss two of them extensively in §3.

## 3 Estimating Concept Effects with CEBaB

We now define the core metrics that we use to evaluate different explanation methods. Figure 1 provides a high-level view of the causal process we are envisioning. The process begins with an exogenous variable $U$ representing a state of the world. For CEBaB, we can imagine that the value of $U$ is a state of affairs $u$ of a person evaluating a restaurant in a particular way. $u$ contributes to a review variable $X$, with the value $x$ of $X$ mediated by $u$ and by mediating concepts $C_1, \ldots C_k$, which correspond to the four aspect-level categories in CEBaB (food, service, ambiance, and noise), each of which can have values $c \in \{\text{positive}, \text{negative}, \text{unknown}\}$. The review $x$ is processed by a model that outputs a vector of scores over classes (sentiment labels in CEBaB).

**Core Metric**  Our central goal is to use CEBaB to evaluate explanation methods themselves. CEBaB supports many approaches to such evaluation. In this paper, we adopt an approach based on individual-level rather than average effects. This makes very rich use of the counterfactual text and associated labels provided by CEBaB. The starting point for this metric is the Individual Causal Concept Effect:

**Definition 1** (Individual Causal Concept Effect; ICaCE). *For a neural network $\mathcal{N}$ and feature function $\phi$, the individual causal concept effect of changing the value of concept $C$ from $c$ to $c'$ for*

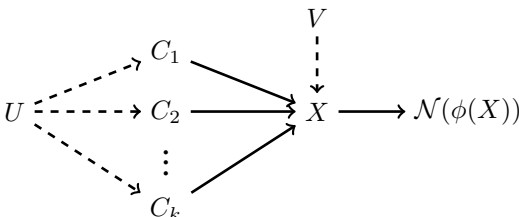

Figure 1: A causal graph describing a data generating process with an exogenous variables $U$ and $V$ representing the state of the world, mediating concepts $C_1, C_2 \ldots, C_k$, and data $X$ that is featurized with $\phi$. $\phi(X)$ is input to a classifier $\mathcal{N}$, which outputs a vector of scores over $m$ output classes.

*state of affairs $u$ in an underlying data generation process $\mathcal{G}$ is*

$$ICaCE_{\mathcal{N}_\phi}(\mathcal{G}, x_u^{C=c}, c') = \mathbb{E}_{x \sim \mathcal{G}} \left[ \mathcal{N}\big(\phi(x)\big) \,\big|\, do\begin{pmatrix} C = c' \\ U = u \end{pmatrix} \right] - \mathcal{N}\big(\phi(x_u^{C=c})\big). \tag{1}$$

ICaCE is a theoretical quantity. In practice, we use the Empirical Individual Causal Concept Effect.

**Definition 2** (Empirical Individual Causal Concept Effect; $\widehat{\text{ICaCE}}$). *For a neural network $\mathcal{N}$ and feature function $\phi$, the empirical individual causal concept effect of changing the value of concept $C$ from $c$ to $c'$ for state of affairs $u$ is*

$$\widehat{ICaCE}_{\mathcal{N}_\phi}(x_u^{C=c}, x_u^{C=c'}) = \mathcal{N}\big(\phi(x_u^{C=c'})\big) - \mathcal{N}\big(\phi(x_u^{C=c})\big), \tag{2}$$

*where $(x_u^{C=c}, x_u^{C=c'})$ is a tuple of inputs originating from $u$ with the concept $C$ set to the values $c$ and $c'$, respectively.*

The $\widehat{\text{ICaCE}}_{\mathcal{N}_\phi}$ for a pair of examples $(x_u^{C=c}, x_u^{C=c'})$ is simply the difference between the output score vectors for the two cases. With CEBaB, we can easily calculate these values because we have clusters of examples that are tied to the same reviewing situation $u$ and express different concept values.

For assessing an explanation method $\mathcal{E}$, we compare ICaCE values with those returned by $\mathcal{E}$. Our core metric is the ICaCE-Error:

**Definition 3** (ICaCE-Error). *For a neural network $\mathcal{N}$, feature function $\phi$ and distance metric Dist, the ICaCE-Error of an explanation method $\mathcal{E}$ for changing the value of concept $C$ from $c$ to $c'$ is:*

$$ICaCE\text{-}Error_{\mathcal{N}_\phi}^{\mathcal{D}}(\mathcal{E}) = \frac{1}{|\mathcal{D}|} \sum_{(x_u^{C=c}, x_u^{C=c'}) \in \mathcal{D}} \text{Dist}\big(\widehat{ICaCE}_{\mathcal{N}_\phi}(x_u^{C=c}, x_u^{C=c'}), \mathcal{E}_{\mathcal{N}_\phi}(x_u^{C=c}, c')\big) \tag{3}$$

We present results for three choices of Dist which vary in their ability to model the direction and magnitude of effects. These choices give subtly different but largely converging results, as detailed in Section 6 and reported more fully in Appendix D.

**Aggregating Individual Causal Concept Effect**   It is often useful to also have a direct estimate of a model's ability to capture concept-level causal effects. For this, we employ an aggregating version of $\widehat{\text{ICaCE}}$, the Empirical Causal Concept Effect:

**Definition 4** (Empirical Causal Concept Effect; $\widehat{\text{CaCE}}$). *For a neural network $\mathcal{N}$ and feature function $\phi$, the empirical causal concept effect of changing the value of concept $C$ from $c$ to $c'$ in dataset $\mathcal{D}$ is*

$$\widehat{CaCE}_{\mathcal{N}_\phi}^{\mathcal{D}}(C, c, c') = \frac{1}{|\mathcal{D}_C^{c \to c'}|} \sum_{(x_u^{C=c}, x_u^{C=c'}) \in \mathcal{D}_C^{c \to c'}} \widehat{ICaCE}_{\mathcal{N}_\phi}(x_u^{C=c}, x_u^{C=c'}). \tag{4}$$

This is an empirical estimator of the Causal Concept Effect (CaCE) of Goyal et al. [17]. It estimates, in general, how the classifier predictions change for a given concept and intervention direction.

Table 2: The evaluated explanation methods and their attributes. **Explainer Method** denotes the complexity of the models used by each explanation method. **Access to Explained Model** denotes the degree of access an explainer method needs to the explained model. **Concept Labels Needed** indicates whether a method estimating the effect for an input $x_u^{C=c}$ needs the actual input label $c$ and/or the intervened value $c'$ at test time. Models with a **Counterfactual Representation** approximate $\phi(x_u^{C=c'})$ to estimate the effect. Finally, only CausaLM and S-Learner have **Confounder Control** to minimize the impact of confounding concepts. *We predict these labels with a classifier.

| Explanation method | Explainer Method | Access to Explained Model | Concept Labels Needed (test time) | Counterfactual Representation | Confounder Control |
|---|---|---|---|---|---|
| Approx | None | None | All concepts and their labels* | ✗ | ✗ |
| CONEXP [17] | None | None | $c$ and $c'$ | ✗ | ✗ |
| S-Learner [27] | Linear | None | All concepts and their labels* | ✗ | ✓ |
| TCAV [26] | Linear | Weights | None | ✗ | ✗ |
| ConceptSHAP [57] | Linear | Weights | None | ✗ | ✗ |
| INLP [40] | Linear | Weights | None | ✓ | ✗ |
| CausaLM [11] | Explained Model | Training Regime | None | ✓ | ✓ |

**Estimating Real-World Causal Effect of Aspect Sentiment on Overall Sentiment**   We can also estimate ground truth causal effects in CEBaB by simply using its labels directly. There are again a variety of ways that this could be done. We opt for the one that makes the richest use of the structures afforded by CEBaB. For perspicuity, in parallel to the neural network-based $\widehat{\text{ICaCE}}$ (Definition 2), we define the Empirical Individual Treatment Effect for our dataset:

**Definition 5** (Empirical Individual Treatment Effects in CEBaB; $\widehat{\text{ITE}}$). *The empirical individual treatment effect of changing the value of concept $C$ from $c$ to $c'$ in CEBaB is*

$$\widehat{\text{ITE}}^{CEBaB}(x_u^{C=c}, x_u^{C=c'}) = f(x_u^{C=c'}) - f(x_u^{C=c}) \tag{5}$$

*where $f$ is a simple look-up procedure that retrieves the overall sentiment labels for CEBaB examples.*

We aggregate over these values by taking their average, in parallel to what we do for network predictions (Definition 4). This yields the Empirical Average Treatment Effect ($\widehat{\text{ATE}}$) for CEBaB.

**Alternative Metrics**   In Appendix A in our supplementary materials, we consider alternative formulations of the core metrics with *causal concept effects* and *absolute causal concept effects*, relating them to the different questions they engage with. We opt for the individual causal concept effect in our central metric (Definition 3), taking the central question to be what caused an ML model to produce an output for an *actual* input created from a real-world process.

## 4   Evaluated Explanation Methods

We compare several model explanation methods that share three main characteristics. First, they are all suitable for NLP models and have been used in the literature for generating model explanations in the form of estimated effects on model predictions. Second, they all provide concept-level explanations, for a pre-defined list of human-interpretable concepts (e.g., how sensitive a restaurant review rating classifier is to language related to food quality). This approach is also forward-looking, allowing more researchers to construct new hypotheses (i.e., concepts we have not collected labels for) and estimate their effect on the predictor. Third, all of the tested methods are model-agnostic, meaning that they separate the explanation from the model. At the same time, these methods differ in five important ways, as summarized Table 2.

We now turn to reviewing the explanation methods that we later compare on CEBaB (§6). In our mathematical formulas, we employ a unified notation for all methods, to make the definitions more accessible and easier to integrate into our experimental set-up. Assume we have a classifier $\mathcal{N}$ (which outputs a probability vector) and feature function $\phi$, and we want to compute the effect on $\mathcal{N}_\phi(x_u^{C=c})$ of changing the value of concept $C$ from $c$ to $c'$ using an unseen test set $(\mathcal{D}, Y)$.

**Approximate Counterfactuals**   The gold labels of CEBaB are the difference between the logits for some original review $x_u^{C=c}$ and ground-truth counterfactual $x_u^{C=c'}$. As a baseline, we sample

an original review $x_{u'}^{C=c'}$ with the same aspect-labels as the $x_u^{C=c'}$ and use it as an approximate counterfactual:

$$\text{Approx}_{\mathcal{N}_\phi}(C, c, c'; x) = \mathcal{N}(\phi(x_{u'}^{C=c'})) - \mathcal{N}(\phi(x_u^{C=c})) \tag{6}$$

We do this sampling using predicted aspect labels from the aspect-level sentiment analysis models described in Appendix C.

**Conditional Expectation (CONEXP)**  Goyal et al. [17] propose a baseline where the effect of a concept $C$ is the average difference in predictions on examples with different values of $C$.

$$\text{CONEXP}_{\mathcal{N}_\phi}^{\mathcal{D}}(C, c, c') = \frac{1}{|\mathcal{D}^{C=c'}|} \sum_{x \in \mathcal{D}^{C=c'}} \mathcal{N}(\phi(x)) - \frac{1}{|\mathcal{D}^{C=c}|} \sum_{x \in \mathcal{D}^{C=c}} \mathcal{N}(\phi(x)) \tag{7}$$

where $\mathcal{D}^{C=c}$ and $\mathcal{D}^{C=c'}$ are subsets of $\mathcal{D}$ where $C$ takes values $c$ and $c'$, respectively. To predict an effect, this method only relies on $C$, $c$, and $c'$, resulting in an estimate that does not depend on the specific input text itself.

**Conditional Expectation Learner (S-Learner)**  We adapt *S-Learner*, a popular method for estimating the Conditional Average Treatment Effect (CATE) [27]. To estimate causal concept effects, our *S-Learner* trains a logistic regression model $\mathcal{E}$ to predict $\mathcal{N}(\phi(x))$ using the values of all the labeled concepts of example $x$, denoted by $x'$.[2] Then, during inference, we compute an individual effect for example pair $(x_u^{C=c}, x_u^{C=c'})$ by comparing the output of the model $\mathcal{E}_x$ on this pair:

$$\textit{S-Learner}(C, c, c'; x) = \mathcal{E}(x'^{C=c'}_u) - \mathcal{E}(x'^{C=c}_u) \tag{8}$$

At inference time, S-Learner assumes access to all aspect-level labels $x'$, which might not always be available. To alleviate this issue, we instead *predict* the aspect-level labels $x'$ from the original text $x$ using models described in Appendix C.

**TCAV**  Kim et al. [26] use *Concept Activation Vectors* (CAVs), which are semantically meaningful directions in the embedding space of $\phi$. Our adapted version of Testing with CAVs (TCAV) outputs a vector measuring the sensitivity of each output class $k$ to changes towards the direction of a concept $v_C$ at the point of the embedded input. It is computed as:

$$\text{TCAV}_{\mathcal{N}_\phi}(C; x) = \left( \nabla \mathcal{N}_k(\phi(x)) \cdot v_C \right)_{k=1}^{K} \tag{9}$$

where $K$ is the number of classes and $v_C$ is a linear separator learned to separate concept $C$ in the embedding space of $\phi$.

**ConceptSHAP**  Yeh et al. [57] propose this expansion to SHAP [30], to generate concept-based explanation based on Shapley values [45]. Given a *complete* (i.e., such that the accuracy it achieves on a test set is higher than some threshold $\beta$) set of $m$ concepts $\{C_1, \ldots, C_m\}$, ConceptSHAP calculates the contribution of each concept to the final prediction. Our adapted version outputs a vector for each $C \in \{C_1, \ldots, C_m\}$ and $x$. We justify this modification and provide implementation details in Appendix H.

**CausaLM**  Feder et al. [11] estimate the causal effect of a binary concept $C$ on the model's predictions by adding auxiliary adversarial tasks to the language representation model in order to learn a counterfactual representation $\phi_C^{\text{CF}}(x)$, while keeping essential information about potential confounders (control concepts). Their method outputs the text representation-based individual treatment effect (TReITE), which is computed as:

$$\text{TReITE}_{\mathcal{N}_\phi}(C; x) = \mathcal{N}'\big(\phi_C^{\text{CF}}(x)\big) - \mathcal{N}\big(\phi(x)\big) \tag{10}$$

where $\phi_C^{\text{CF}}$ denotes the learned counterfactual representation, where the information about concept $C$ is not present, and $\mathcal{N}'$ is a classifier trained on this counterfactual representation. A key feature of CausaLM is its ability to control for confounding concepts (if modeled).[3] An inherent drawback of this technique is that it can only estimate interventions well for $c' = $ Unknown, since the counterfactual representation is only trained to *remove* a concept $C$.

---

[2]This training approach, where an explainer model is fit to predict the output of the original model, shares the intuition of LIME, the widely used explanation method [41], but for concept-level effects.

[3]As in Feder et al. [11], we control for the most correlated potential confounder.

Table 3: Dataset statistics of CEBaB combining train/dev/test splits.

| | Positive | Negative | Unknown | no maj. | Total |
|---|---|---|---|---|---|
| food | 5726 (41%) | 5526 (38%) | 2605 (15%) | 208 (31%) | 14065 |
| service | 4045 (29%) | 4098 (28%) | 3877 (22%) | 178 (27%) | 12198 |
| ambiance | 2928 (21%) | 2597 (18%) | 5121 (29%) | 203 (30%) | 10849 |
| noise | 1365 (10%) | 2215 (15%) | 5883 (34%) | 78 (12%) | 9541 |

(a) Aspect-level labels.

| 1 star | 1870 (12%) |
|---|---|
| 2 star | 3056 (20%) |
| 3 star | 3517 (23%) |
| 4 star | 2035 (13%) |
| 5 star | 2732 (18%) |
| no maj. | 1879 (12%) |

(b) Review-level ratings.

| | {Neg, Pos} | {Neg, Unk} | {Pos, Unk} |
|---|---|---|---|
| food | 898 | 1316 | 1291 |
| service | 851 | 857 | 938 |
| ambiance | 947 | 585 | 472 |
| noise | 1145 | 208 | 260 |

(c) Edit pair distribution. Edit pairs are examples that come from the same original source text and differ only in their rating for a particular aspect.

| | Neg to Pos | Neg to Unk | Pos to Unk |
|---|---|---|---|
| food | 1.84 | 1.37 | −1.02 |
| service | 0.98 | 0.91 | −0.53 |
| ambiance | 0.93 | 0.91 | −0.50 |
| noise | 0.72 | 0.48 | −0.47 |

(d) Empirical $\widehat{\text{ATE}}$ for the five-way sentiment labels in CEBaB. The reverse of a given concept change is the negative of the value given – e.g., the $\widehat{\text{ATE}}$ for 'Pos to Neg' for food is −1.84. See Appendix B for the corresponding values for binary sentiment.

**Iterative Nullspace Projection (INLP)** Ravfogel et al. [40] remove a concept from a representation vector by repeatedly training linear classifiers that aim to predict that attribute from the representations and projecting the learned representations on their null-space. Similar to CausaLM, INLP also estimates the TReATE (Equation 10) and can only estimate interventions for $c' =$ Unknown.

## 5 The CEBaB Dataset

Table 1 provides an intuitive overview of the structure of CEBaB. In the *editing* phase of dataset creation, crowdworkers modified an existing OpenTable review in an effort to achieve a specific aspect-level goal while holding all other properties of the original text constant. Our aspect-level categories are food, ambiance, service, and noise. In the *validation* phrase, crowdworkers labeled each example relative to each aspect as 'Positive', 'Negative', or 'Can't tell' (Unknown). Having five labels per example allows us to infer a majority label or reason in terms of the full label distributions. In the *rating* phase, each full text was labeled using a common five-star scale, again by five crowdworkers.

We began with 2,299 original reviews from OpenTable (related to 1,084 restaurants) and expanded them, via the above editing procedure, into a total of 15,089 texts. The distribution of normalized edit distances has peaks around 0.28 and 0.77, showing that workers made non-trivial changes to the originals, and even often had to make substantial changes to achieve the editing goal. (See Appendix B for the full distribution.)

Table 3 summarizes the resulting label distributions, where an example has label $y$ if at least 3 of the 5 labelers chose $y$, otherwise it is in the 'no majority' category. 99% of aspect-level edits have a majority label that corresponds to the editing goal, and 88% of the texts have a review-level majority label on the five-star scale. Overall, these percentages show that workers were extremely successful in achieving their editing goals and that edits have systematic effects on overall sentiment.

The central goal of CEBaB is to create *edit pairs*: pairs of examples that come from the same original text and differ only in their labels for a particular aspect. For example, in Table 1, the first two 'food edit' cases form an edit pair, since they come from the same original text and differ only in their food label. Original texts can also contribute to edit pairs; the original text in Table 1 forms an edit pair with each of the texts it is related to by edits. Table 3c summarizes the distribution of edit pairs, and Table 3d reports the ground-truth $\widehat{\text{ATE}}$ values (§3).

Table 4: $\widehat{\mathrm{CaCE}}$ (Definition 4) for `bert-base-uncased` fine-tuned as a 5-way sentiment classifier. Rows are concepts, columns are real-world concept interventions, and each entry indicates the average change in classifier output when the concept is intervened on with the given direction.[4] Results are averaged over 5 distinct seeds with standard deviations. The $\widehat{\mathrm{CaCE}}$ value of changing concept $C$ from $c$ to $c'$ is the negative $\widehat{\mathrm{CaCE}}$ value of changing concept $C$ from $c'$ to $c$.

|  | Negative to Positive | Negative to unknown | Positive to unknown |
|---|---|---|---|
| food | 1.90 (± 0.03) | 1.00 (± 0.02) | −0.82 (± 0.01) |
| service | 1.42 (± 0.04) | 0.89 (± 0.04) | −0.45 (± 0.01) |
| ambiance | 1.27 (± 0.01) | 0.79 (± 0.01) | −0.50 (± 0.03) |
| noise | 0.75 (± 0.02) | 0.44 (± 0.00) | −0.23 (± 0.02) |

We release the dataset with fixed train/dev/test splits. In creating these splits, we enforce two high-level constraints. The first is our 'grouped' requirement: for each original review $t$, all texts that are related to $t$ via editing occur in the same split as $t$. This ensures that models are not evaluated on examples that are related by editing to those they have seen in training. Second, if any text $t$ in a group received a 'no majority' label, then the entire group containing $t$ is put in the train set. This ensures that there is no ambiguity about how to evaluate models on dev and test examples.

Once these high-level conditions were imposed, the examples were sampled randomly to create the splits. This allows that individual workers can contribute edited texts across splits. This minor compromise was necessary to ensure that we could have large dev and test splits. Appendix C in our supplementary materials shows that worker identity has negligible predictive power.

There are two versions of the train set: *inclusive* and *exclusive*. The inclusive train set contains all original and edited non-dev/test texts (11,728 texts). The exclusive version samples exactly one train text from each set of texts that are related by editing (1,755 examples). The rationale is that models trained with an original review as well as its edited counterparts may explicitly learn causal effects trivially by aggregating learning signals across inputs. Our exclusive train split prevents this, which helps facilitate fair comparisons between explanation methods and better resembles a real-world setting.

Our dataset is released publicly in JSON format and is available in the Hugging Face `datasets` library. It includes restaurant metadata, full rating distributions, and anonymized worker ids. Appendix B in our supplementary materials provides additional details on the dataset construction, including the prompts used by the crowdworkers, the number of workers per task, worker compensation, and a sample of examples with ratings to help convey the nature of workers' edits and the overall quality of the resulting texts and labels. In addition, Appendix C reports on a wide range of classifier experiments at the aspect-level and text-level that show that models perform well on CEBaB classification tasks, which bolsters the claim that CEBaB is a reliable tool for assessing explanation methods.

## 6 Experiments and Results

For each experiment, we fine-tune a pretrained language model to predict the overall sentiment of all restaurant reviews from our *exclusive* OpenTable train set. Since the goal of our work is not to achieve state-of-the-art performance, but rather to compare explanation methods and demonstrate the usage of CEBaB, we test the ability of methods to explain commonly used models, trained with standard experimental configurations.

In the main text, we report results for `bert-base-uncased` fine-tuned as a five-way classifier. Appendix D includes results for `GPT-2`, `RoBERTa`, and an `LSTM`, fine-tuned on binary, 3-way and 5-way versions of the sentiment task. All results, including the ground-truth effect that depends on the specific instance of a model, are averaged across 5 seeds.

To evaluate the intrinsic capacity of a model to capture causal effects, we report the $\widehat{\mathrm{CaCE}}$ values, as in Definition 4. The results for `bert-base-uncased` are given in Table 4. They are intuitive and

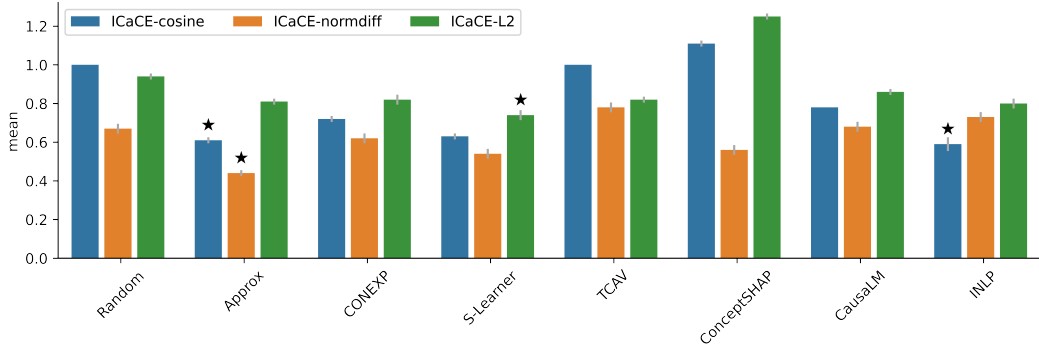

Figure 2: ICaCE-Error (Definition 3) for `bert-base-uncased` fine-tuned for five-way sentiment, averaged per aspect. We report values for cosine, L2, and normdiff. **Lower is better**. Stars mark the best result(s) per metric. Results averaged over 5 distinct seeds. [†]RandomExplainer takes the difference between two random probability vectors as the predicted effect.

well-aligned with the $\widehat{\text{ATE}}$ estimates in Table 3d, indicating that the model has captured the real-world effects.

Our primary assessment of the evaluation methods is given in Figure 2, again focusing on a five-way `bert-base-uncased` model as representative of our results. We provide values based on *cosine*, *L2*, and *normdiff* as the value of Dist in Definition 3. The *cosine*-distance metric measures if the estimated and observed effect have the same direction but does not take the magnitudes of the effects into account. The *L2*-distance measures the Euclidian norm of the difference of the observed and estimated effect. Both the direction and magnitude of the effects influence this metric. To only compare the magnitudes, we use the *normdiff*-distance, which computes the absolute difference between the Euclidean norms of the observed and estimated effects, thus completely ignoring the directions of both effects.

Remarkably, our approximate counterfactual baseline proves to be the best method at capturing both the direction and magnitude of the effects. The fact that a simple baseline method beats almost all other methods indicates that we need better explanation methods if we are going to capture even relatively simple causal effects like those given by CEBaB.

Recall from Table 2 that the compared methods require different levels of access to concept labels at inference time. Approximate counterfactuals and S-Learner have access to both the direction of the intervention and the predicted test-time aspect labels, enabling them to outperform CONEXP, which has access to only the direction of the intervention, and TCAV, ConceptSHAP, and CausaLM, which have access to neither the intervention direction nor test-time aspect labels.

The INLP method ties with the best method for the *cosine* metric, despite having access to neither intervention directions nor test-time aspect labels. Perhaps this method could be extended to make use of this additional information and decisively improve upon our approximate counterfactual baseline.

While CausaLM and INLP both estimate the effect of removing a concept from an input, INLP uses linear probes to guide interventions on the original model, while CausaLM trains an entirely new model with an auxiliary adversarial objective. The direct use of the original model is something INLP shares with the approximate counterfactual baseline; it seems that a tight connection to the original model may underlie success on CEBaB.

## 7 Conclusion

Our main contributions in this paper are twofold. First, we introduced CEBaB, the first benchmark dataset to support comparing different explanation methods against a single ground-truth with human-created counterfactual texts and multiply-validated concept labels for aspect-level and overall

---

[4]Definition 4 defines the CaCE values as vectors. In this table, we collapse the CaCE values to scalars by having $\mathcal{N}$ output the most probable predicted class, instead of the class distribution.

sentiment. Using this resource, one can isolate the true causal concept effect of aspect-level sentiment on any trained overall sentiment classifier. CEBaB provides a level playing field on which we can compare a variety of explanation methods that differ in their assumptions about their access to the model, their computational demands, their access to ground-truth concept labels at inference time, and their overall conception of the explanation problem. Furthermore, the evaluated methods make absolutely no use of CEBaB's counterfactual train set. In turn, we hope that CEBaB will facilitate the development of explanation methods that can take advantage of the very rich counterfactual structure CEBaB provides across all its splits.

Second, we have provided an in-depth experimental analysis of how well multiple model explanation methods are able to capture the true concept effect. A naive baseline that approximates counterfactuals through sampling achieves the best performance, with INLP and S-Learner being the only other methods that achieves state-of-the art on any metric. While CEBaB is only grounded in one task, sentiment analysis alone is enough to produce starkly negative results that should serve as a call to action for NLP researchers aiming to explain their models.

## Acknowledgments and Disclosure of Funding

This research is supported in part by a grant from Meta AI. Karel D'Oosterlinck was supported through a doctoral fellowship from the Special Research Fund (BOF) of Ghent University. We thank our crowdworkers for their invaluable contributions to CEBaB.

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
