## Supplementary Materials

## A    Causal Concept Effects and Metrics for Explanation Methods

Data do not materialize out of thin air. Rather, data are generated from real-world processes with complex causal structures we do not observe directly. Causal inference is the task of estimating theoretical causal effect quantities.

When estimating causal effects, researchers commonly measure the *average treatment effect*, which is the difference in mean outcomes between the treatment and control groups [42]. Formally, we define the average treatment effect of binary treatment $T$ on an outcome $Y$ under a data generation process $\mathcal{G}$ that represents the unknown details of the real-world.

**Definition 6** (Average Treatment Effect; ATE [42,35])**.**

$$ATE_T(Y, \mathcal{G}) = \mathbb{E}_{\mathcal{G}}\big[Y \,\big|\, do(T = 1)\big] - \mathbb{E}_{\mathcal{G}}\big[Y \,\big|\, do(T = 0)\big]. \tag{11}$$

The ATE is a theoretical quantity we cannot compute in practice, since we do not have access to $\mathcal{G}$ nor can we observe both interventions for the same subject.

However, we are concerned with estimating the causal effect of variables representing *non-binary concepts* in real-world systems, on data in an appropriate format for processing by a modern AI model that predicts *vector encoding probability distributions* over outputs.

Let $\mathcal{N}$ be a neural network outputting a probability vector, where its $k$-th entry represents the probability to predict the $k$-th class, and let $\phi$ be a feature representation (e.g., BERT embedding). In the context of model explanations, we will define the tools needed to answer three questions:

1. Given a real-world circumstance $u$ that led to input data $x_u^{C=c}$, what is expected effect of a concept $C$ changing from value $c$ to value $c'$ on the model output of $\mathcal{N}_\phi$ provided input data $x_u^{C=c}$?

2. What is the expected effect of a concept $C$ changing from value $c$ to value $c'$ on the output of the model $\mathcal{N}_\phi$ provided input data $X$ across real-world circumstances $U$?

3. What is the magnitude of the expected effect of a changing the concept $C$ on the output of the model $\mathcal{N}_\phi$ provided input data $X$ across real-world settings $U$?

For example, in the context of CEBaB, we might ask

1. Given a real-world dining experience $u$ with good food quality ($C_{\text{food}} = +$) that led to a restaurant review $x_u^{C_{\text{food}}=+}$, what is the effect of changing the food quality $C_{\text{food}}$ from $C_{\text{food}} = +$ to $C_{\text{food}} = -$ on the output of an overall-sentiment text classifier $\mathcal{N}_\phi$ provided a review of the dining experience?

2. What is the expected effect of changing the food quality $C_{\text{food}}$ from positive $+$ to negative $-$ on the output of the model $\mathcal{N}_\phi$ across real-world dining experiences that lead to restaurant reviews?

3. What is the magnitude of the expected effect of a changing food quality $C_{\text{food}}$ on the output of the model $\mathcal{N}_\phi$ across real-world dining experiences that lead to restaurant reviews?

Each of the above questions requires the estimation of a different theoretical quantity. In respect to the order of the questions, these quantities are the *individual causal concept effect*, the *causal concept effect*, and the *absolute causal concept effect*.

We believe the most practical question in explainable AI is: why does this model have this output behavior for an *actual* input. For this reason, our focus in the main text is *individual causal concept effects*. We define our central metric that captures the performance of an explainer on CEBaB as the average error on individual causal effect predictions (Definition 3).

We do not evaluate the ability of explainers to evaluate the causal concept effect or the absolute causal concept effect.

### A.1 Theoretical Quantities

**Definition 7** (Causal Concept Effects; [17]). *For an exogenous setting $u$ that led to concept $C$ taking on value $c$ and the creation of input data $x_u^{C=c}$, the individual causal concept effect of a concept $C$ changing from value $c$ to $c'$ in a data generation process $\mathcal{G}$ on a neural network $\mathcal{N}$ with feature representation $\phi$ is*

$$ICaCE_{\mathcal{N}_\phi}(\mathcal{G}, x_u^{C=c}, c') = \mathbb{E}_{x \sim \mathcal{G}} \left[ \mathcal{N}(\phi(x)) \,\middle|\, do\begin{pmatrix} C = c' \\ U = u \end{pmatrix} \right] - \mathcal{N}(\phi(x_u^{C=c})) \tag{12}$$

*The causal concept effect is the effect in general, meaning there is no input data generated from a fixed exogenous real-world setting:*

$$CaCE_{\mathcal{N}_\phi}(\mathcal{G}, C, c, c') = \mathbb{E}_{x \sim \mathcal{G}}\big[\mathcal{N}(\phi(x)) \,\big|\, do(C = c')\big] - \mathbb{E}_{x \sim \mathcal{G}}\big[\mathcal{N}(\phi(x)) \,\big|\, do(C = c)\big] \tag{13}$$

*The absolute causal concept effect estimate of the magnitude of the effect a concept has on a classifier output, regardless the concept values. We aggregate over all possible intervention values in the following way*

$$ACaCE_{\mathcal{N}_\phi}(\mathcal{G}, C) = \frac{1}{|\{\{c, c'\} \subseteq C\}|} \sum_{\{c,c'\} \subseteq C} \big|CaCE_{\mathcal{N}_\phi}(\mathcal{G}, C, c, c')\big|, \tag{14}$$

*where $C$ is the set of all possible values for concept in addition to denoting the concept itself.*[5]

### A.2 Empirical Estimates

Similar to the ATE, causal concept effects are theoretical quantities we can only estimate in reality. To perform such estimates, we need a dataset consisting of pairs $(x_u^c, x_u^{c'}) \in \mathcal{D}$ that are drawn from a data generation process $\mathcal{G}$. A major contribution of this work is crowdsourcing such a dataset, CEBaB. These pairs allow us to compute empirical estimations of (individual) causal concept effects.

**Definition 8** (Empirical Causal Concept Effects). *For an exogenous setting $u$, the empirical individual causal concept effect of a concept $C$ changed from value $c$ to $c'$, for $\mathcal{D}$ sampled from $\mathcal{G}$, on a neural network $\mathcal{N}$ trained on a feature representation $\phi$ is*

$$\widehat{ICaCE}_{\mathcal{N}_\phi}(x_u^{C=c'}, x_u^{C=c}) = \mathcal{N}(\phi(x_u^{C=c'})) - \mathcal{N}(\phi(x_u^{C=c})) \tag{15}$$

*Given a full dataset $\mathcal{D}$ of such pairs, we can estimate the causal concept effect*

$$\widehat{CaCE}_{\mathcal{N}_\phi}(\mathcal{D}, C, c, c') = \frac{1}{|\mathcal{D}_C^{c \to c'}|} \sum_{(x_u^c, x_u^{c'}) \in \mathcal{D}} \widehat{ICaCE}_{\mathcal{N}_\phi}(x_u^{C=c}, x_u^{C=c'}) \tag{16}$$

*And also the absolute causal concept effect*

$$\widehat{ACaCE}_{\mathcal{N}_\phi}(\mathcal{D}) = \frac{1}{|\{\{c, c'\} \subseteq C\}|} \sum_{(c,c') \in C} |\widehat{CaCE}_{\mathcal{N}_\phi}(\mathcal{D}, C, c, c')| \tag{17}$$

Notice that the only difference between causal concept effects (Definition 7) and empirical causal concept effects (Definition 8) is that we change the expectation taken over $\mathcal{G}$ to be the average over a dataset $\mathcal{D} \sim \mathcal{G}$.

### A.3 Explainer Errors

Given a dataset $\mathcal{D}$ and an explainer $\mathcal{E}_{\mathcal{N}_\phi}(x_u^c, c')$ that predicts individual causal concept effects $ICACE_{\mathcal{N}_\phi}(x_u^c, c')$, we define metrics capturing the ability of $\mathcal{E}$ to estimate causal effects by simple computing the averaged distance between our explainer and the empirical causal effect

---

[5]We take the absolute value since $CaCE_{\mathcal{N}_\phi}(\mathcal{G}, C, c, c') = -CaCE_{\mathcal{N}_\phi}(\mathcal{G}, C, c', c)$, and these cancel each other in the summation.

**Definition 9** (Explainer Distances). *The average distance between the explainer and the empirical individual causal concept effects.*

$$ICaCE\text{-}Error^{\mathcal{D}}_{\mathcal{N}_\phi}(\mathcal{E}, C, c, c') =$$
$$\frac{1}{\left|\mathcal{D}_C^{c \to c'}\right|} \sum_{(x_u^{C=c}, x_u^{C=c'}) \in \mathcal{D}_C^{c \to c'}} \mathsf{Dist}\left(\widehat{ICaCE}_{\mathcal{N}_\phi}(x_u^{C=c}, x_u^{C=c'}), \mathcal{E}_{\mathcal{N}_\phi}(x_u^{C=c}, x_u^{C=c'})\right) \quad (18)$$

*The distance between the average of explainer outputs and the empirical causal concept effect*

$$CaCE\text{-}Error^{\mathcal{D}}_{\mathcal{N}_\phi}(\mathcal{E}, C, c, c') = \|\widehat{CaCE}_{\mathcal{N}_\phi}(\mathcal{D}, C, c, c'), \; \frac{1}{\left|\mathcal{D}_C^{c \to c'}\right|} \sum_{x_u^c, x_u^{c'} \in \mathcal{D}_C^{c \to c'}} \mathcal{E}_{\mathcal{N}_\phi}(x_u^c, c'))\| \quad (19)$$

*The distance between the average magnitude of explainer outputs and the empirical absolute causal effect*

$$ACaCE\text{-}Error^{\mathcal{D}}_{\mathcal{N}_\phi}(\mathcal{E}, C) =$$
$$\|\widehat{ACaCE}_{\mathcal{N}_\phi}(\mathcal{D}, C), \; \frac{1}{|\{\{c, c'\} \subseteq C\}|} \sum_{(c, c') \in C} \frac{1}{\left|\mathcal{D}_C^{c \to c'}\right|} \sum_{x_u^c, x_u^{c'} \in \mathcal{D}_C^{c \to c'}} |\mathcal{E}_{\mathcal{N}_\phi}(x_u^c, c'))|\| \quad (20)$$

*where $\| \cdot \|$ is some distance metric and $\mathcal{D}_C$ is the subset of data where $C$ is the concept changed and $\mathcal{D}_C^{c \to c'}$ is the subset of data where $C$ is the concept changed from value $c$ to value $c'$.*

In the main text, we use the ICaCE-Error as our primary evaluation metric.

## B CEBaB

Our supplementary materials contain a full Datasheet for CEBaB as a separate markdown document.

### B.1 Restaurant-level metadata from OpenTable

Table 5 gives an overview of the metadata associated with the original review texts in CEBaB.

Table 5: CEBaB metadata from OpenTable, tabulated at the level of individual original reviews. A total of 1,084 restaurants are represented in the data.

| italian | 1076 |
|---|---|
| american | 654 |
| french | 254 |
| seafood | 202 |
| mediterranean | 113 |

(a) Cuisine.

| northeast | 863 |
|---|---|
| west | 634 |
| south | 470 |
| midwest | 332 |

(b) U.S. regions.

| 1 star | 244 |
|---|---|
| 2 star | 1207 |
| 3 star | 123 |
| 4 star | 330 |
| 5 star | 395 |

(c) Star ratings.

### B.2 Crowdworkers

A total of 254 workers participated in our experiments. All of them come from a pool of workers whom we prequalified to participate in our tasks based on the work they did for us on previous crowdsourcing projects. Thus, we expected that they would do high quality work, and they more than lived up to our expectations, as indicated by the high degree of success they achieved when editing and the high degree of consensus they reached about how to label examples.

There are a total of 642 instances of 15,0006 for which, despite our best efforts, a worker validated an example that they themselves created during the editing phase. Removing the contributions of these workers affects the majority in only 24 cases, with no clear pattern to the changes, so we kept all the validation labels in order to ensure that every example has give responses.

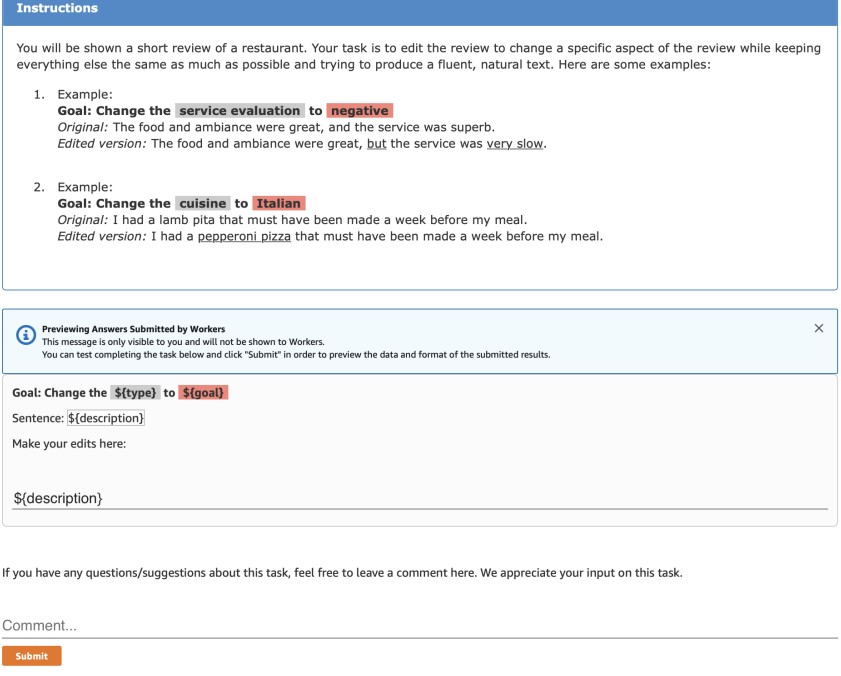

Figure 3: Edit phase annotation interface where the task was to convey 'Positive' or 'Negative' for the target aspect.

### B.3 Editing Phase

A total of 183 workers participated in this phase. Workers were paid US$0.25 per example. Figure 3 shows the annotation interface that workers used when changing the target aspect's sentiment to either 'Positive' or 'Negative', and Figure 4 shows the interface where the task was to hide the target aspect's sentiment.

Figure 5 summarizes the distribution of edit distances between original and edited texts. These distances are calculated at the character-level and normalized by the length of the original or review, whichever is longer.

### B.4 Validation Phase

A total of 174 workers participated in this phase. Workers were paid US$0.35 per batch of 10 examples. Figure 6 shows the annotation interface that workers used.

### B.5 Review-level Rating Phase

A total of 155 workers participated in this phase. Workers were paid US$0.35 per batch of 10 examples. Figure 7 shows the annotation interface that workers used.

### B.6 Randomly Selected Examples

Table 6 provides a random sample of edit pairs from CEBaB's dev set.

### B.7 Five-way Empirical ATE for CEBaB

Table 7 provides the binary $\widehat{\text{ATE}}$ values for CEBaB. These can be compared with the corresponding five-way values in Table 3d in the main text.

Table 6: Randomly sampled edit pairs from CEBaB.

| description | original? | aspect | edit goal | aspect labels | aspect maj. | review labels | review maj. |
|---|---|---|---|---|---|---|---|
| Food was disgusting and very unreasonable!!!!! Every request was honored and very friendly staff.\nHomemade bread which was foul...... | False | food | – | –, –, –, – | – | 2, 2, 2, 2, 2 | 2 |
| Every request was honored and very friendly staff. | False | food | unk. | unk, unk, unk, +, + | unk. | 5, 5, 5, 4, 4 | 5 |
| The food was average, but the service was terrible. | True | food | None | –, –, unk, + | – | 2, 2, 2, 3, 3 | 2 |
| The food was above average, but the service was terrible. | False | food | + | +, +, +, + | + | 3, 3, 3, 3, 2 | 3 |
| We hated our afternoon at Shorebreak! | False | ambiance | – | –, –, unk, unk | – | 1, 1, 1, 1, 1 | 1 |
| We loved our afternoon at Shorebreak! | False | ambiance | unk. | unk, unk, unk, –, + | unk. | 5, 5, 5, 4, 4 | 5 |
| The Sunday Jazz Brunch is great - Good music and fine, creative food. The service was great, my server answered all of my questions. The ambiance is quiet, but not so quiet as to inhibit conversation. A wonderful way to spend an early Sunday afternoon. | False | service | + | +, +, +, +, + | + | 5, 5, 5, 4, 4 | 5 |
| The Sunday Jazz Brunch is great - Good music and fine, creative food. The ambiance is quite, but not so quite as to inhibit conversation. A wonderful way to spend an early Sunday afternoon. The only bad spot was the horrid service. | False | service | – | –, –, –, – | – | 4, 4, 4, 4, 3 | 4 |
| My pasta dish was flavorless and rubbery and my husband's was cold. At least it 45 minutes to get it. Very poor, indeed. | True | food | None | –, –, –, – | – | 1, 1, 1, 2, 2 | 1 |
| My pasta dish was amazing and cooked great. At least it 45 minutes to get it. Very poor, indeed. | False | food | + | +, +, +, +, – | + | 3, 3, 3, 3, 1 | 3 |
| liked the restaurant a lot and loved the meal. Found the chicken great! | False | food | + | +, +, +, +, + | + | 5, 5, 5, 4, 3 | 5 |
| I liked the restaurant a lot, | False | food | unk. | unk, unk, unk, unk, + | unk. | 5, 5, 5, 4, 4 | 5 |
| At the heart of it, this is a HOTEL restaurant. | True | noise | None | unk, unk, unk, unk, unk | unk. | 3, 3, 3, 3, 2 | 3 |
| At the heart of it, this is an extremely loud restaurant. | False | noise | – | –, –, –, – | – | 1, 1, 1, 3, 2 | 1 |
| I was expecting some dishes from the Northern Italian Cuisine. The menu was not distinguishable from any other chain. The food was good but no differentiation. It was noisy, but I believe by design. | True | food | None | +, +, +, +, + | + | 3, 3, 3, 4, 2 | 3 |
| I was expecting some dishes from the Northern Italian Cuisine. The menu was not distinguishable from any other chain. The food was even worse than that. It was also noisy, but I believe by design. | False | food | – | –, –, –, + | – | 1, 1, 1, 2, 2 | 1 |

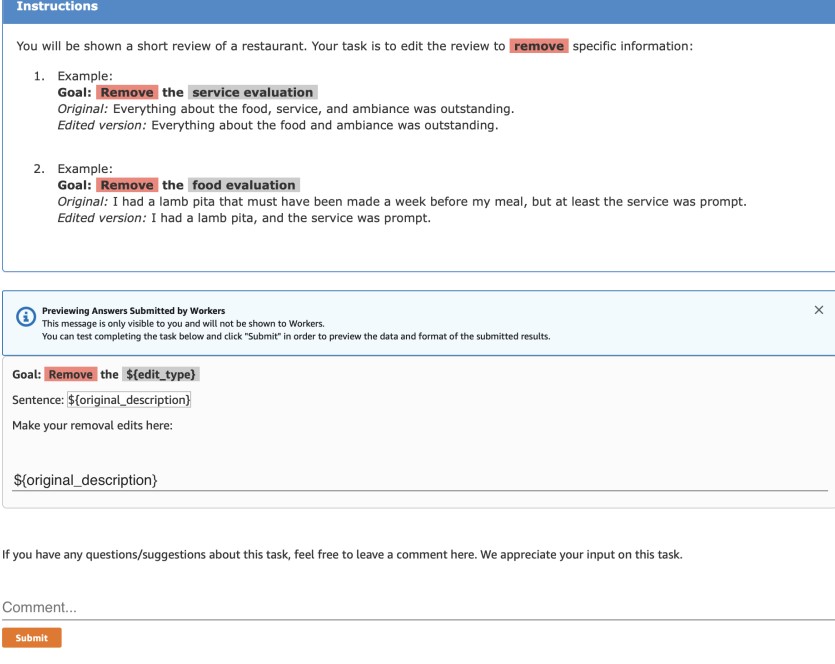

Figure 4: Edit phase annotation interface where the task was to hide the sentiment of the target aspect.

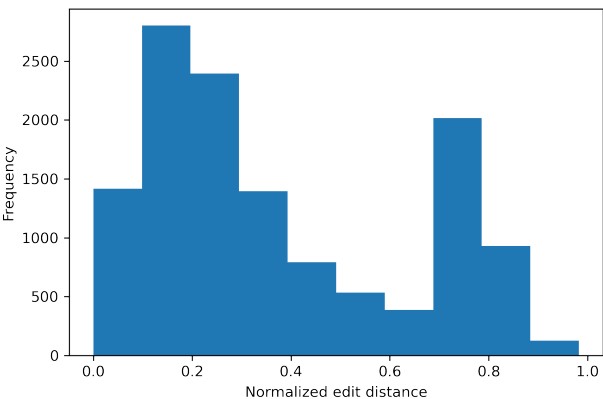

Figure 5: Normalized edit distances between original texts and those created during the editing phase for CEBaB.

### B.8 Edit variability

In the editing phase we ask human annotators to produce edits of an original review with regard to some concept. This is inherently a noisy process, which may impact the quality of our final benchmark. The CEBaB dataset features a modest set of paired edits (176 pairs in total). Each of these pairs contains two edits, starting from the same original sentence and edit goal, which results in two different edited sentences. Like all sentences in CEBaB, these edits were labeled for their review score by human annotators.

Figure 8a shows the distribution of the difference in final review majorities produces by these paired edits. Most paired edits differ at most by one star in their final majority rating, indicating that in general there is some noise associated with the editing procedure, but this does not have a major impact on the final review score. Figure 8b shows the same distribution when we consider the average

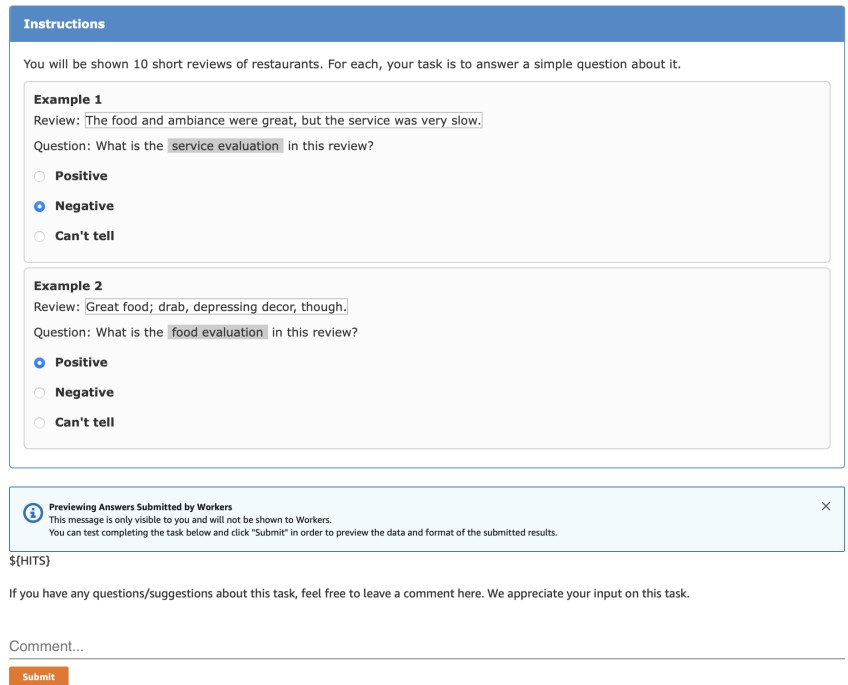

Figure 6: Validation phase annotation interface.

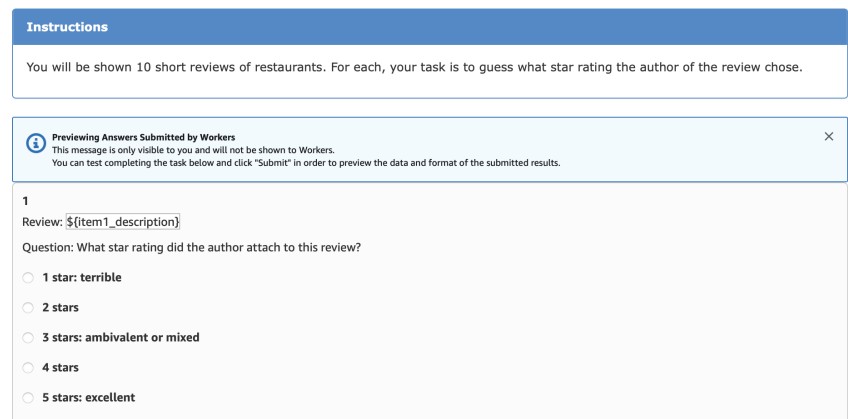

Figure 7: Review-level annotation interface.

Table 7: Empirical $\widehat{\text{ATE}}$ for the binary sentiment labels in CEBaB. Reversing concept order results in the negation of the value given.

|  | Neg to Pos | Neg to Unk | Pos to Unk |
|---|---|---|---|
| food | 0.77 | 0.49 | −0.41 |
| service | 0.25 | 0.20 | −0.16 |
| ambiance | 0.14 | 0.18 | −0.14 |
| noise | 0.08 | 0.04 | −0.14 |

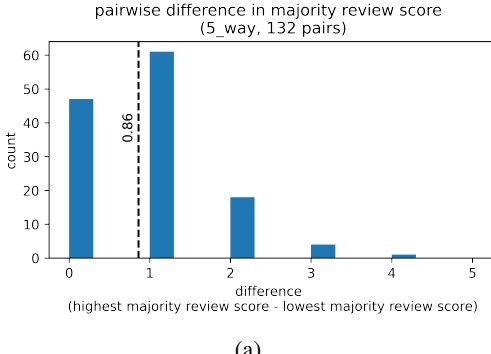
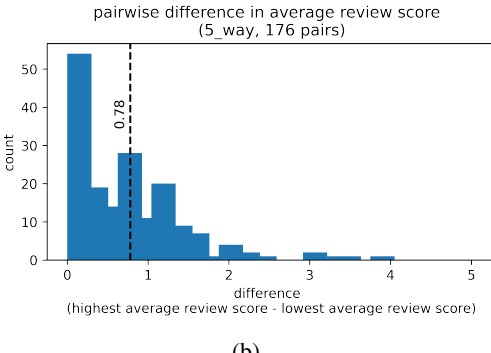

(a)                                                   (b)

Figure 8: Pairwise absolute difference in majority (a) and average (b) review score for all double edits. Figure (a) only considers the 132 pairs where both edits have an actual review majority. Figure (b) considers all 176 pairs. Averages of the distributions are shown with a dotted vertical line.

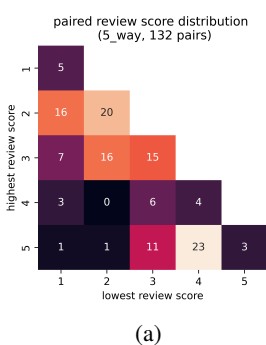
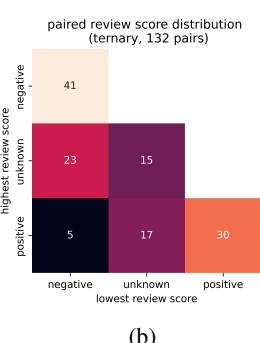
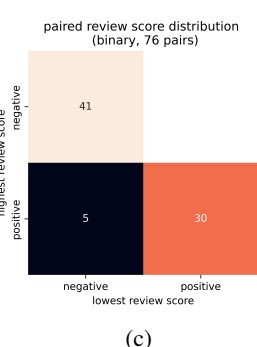

(a)                                    (b)                                    (c)

Figure 9: Pairwise review majority distribution for all double edits in 5-way (a), ternary (b), and binary (c) classification settings. Figures (a) and (b) consider only the 132 pairs where both edits have an actual review majority. Figure (c) considers the 76 pairs that have both a review majority and non-neutral labels.

review score an edit received, as opposed to the majority score. If we consider these average scores, most of the paired edits differ only slightly in their resulting review score.

Figures 9a-c shows the distribution of this pairwise review score in more detail. In an idealized setting without variability, the distribution would be centered around the diagonal of the heatmap. When going from 5-way classification to ternary and binary classification, the variability introduced by the edits becomes less relevant with regard to the final review majority label.

## C    CEBaB Modeling Experiments

This section reports on standard classifier-based experiments with CEBaB, aimed at providing a sense for the dataset when it is used as a standard supervised sentiment dataset. We report experiments on the aspect-level and review-level ratings. In addition, we present evidence that author identity does not have predictive value.

### C.1    Experiments Set-up

We rely on the Hugging Face `transformers` library.[6] [56] We train our models with 4 Nvidia 2080 Ti RTX 11GB GPUs on a single node machine. We use a maximum sequence length of 128 with a fix batch size of 32 with a initial learning rate of $2e^{-5}$. We run each experiment 5 times with distinct random seeds. We train our models with a minimum epoch number of 5 with our largest training set. We linearly scale our training epoch number by the size of the training set. We skip hyperparameter

---

[6]https://github.com/huggingface/transformers

Table 8: Model performance results for sequence classification as well as aspect-based sentiment analysis (ABSA) under 3 training conditions. Mean Macro-F1 scores across 5 runs with distinct random seeds are reported.

| Model | Exclusive | | | | Inclusive | | | |
|---|---|---|---|---|---|---|---|---|
| | Binary | Ternary | 5-way | ABSA | Binary | Ternary | 5-way | ABSA |
| | | | | dev split | | | | |
| BERT | 0.97 | 0.82 | 0.68 | 0.88 | 0.98 | 0.85 | 0.72 | 0.90 |
| GPT-2 | 0.97 | 0.80 | 0.67 | 0.88 | 0.98 | 0.84 | 0.70 | 0.89 |
| LSTM | 0.94 | 0.75 | 0.59 | 0.83 | 0.96 | 0.82 | 0.68 | 0.87 |
| RoBERTa | 0.99 | 0.83 | 0.71 | 0.89 | 0.99 | 0.86 | 0.76 | 0.90 |
| | | | | test split | | | | |
| BERT | 0.97 | 0.82 | 0.70 | 0.87 | 0.98 | 0.84 | 0.73 | 0.89 |
| GPT-2 | 0.97 | 0.80 | 0.65 | 0.87 | 0.97 | 0.83 | 0.68 | 0.89 |
| LSTM | 0.94 | 0.75 | 0.60 | 0.82 | 0.96 | 0.81 | 0.68 | 0.87 |
| RoBERTa | 0.98 | 0.83 | 0.70 | 0.88 | 0.99 | 0.86 | 0.75 | 0.90 |

tuning for optimized task performance as our goal for this paper is to evaluate explanation methods. We release all of our models on Huggingface Dataset Hub.

## C.2  Models

We include 4 different types of models, including BERT (bert-base-uncased) [7], RoBERTa (roberta-base) [29], GPT-2 (gpt2) [38], as well as LSTM with dot-attention [31]. Our LSTM model uses bert-base-uncased tokenizer for simplicity. We initialize the embeddings of tokens for our LSTM using fastText [24]. We reconfigure the classification head all other models the same classification head as in RoBERTa as a non-linear multilayer perceptron (MLP).[7]

## C.3  Multi-class Sentiment Analysis Benchmark

We report model performance results under 3 training conditions: **Binary Classification**, where we label reviews with 1 star and 2 star ratings as negative, reviews with 4 star and 5 star as positive, and 3-star reviews are dropped; **Ternary Classification**, where we add another neutral class for reviews with 3 star ratings; and **5-way Classification**, where each star rating by itself is considered as a class. We leave out reviews in the train set in the 'no majority' category. (Dev and Test do not contain any such examples.) Table 8 shows the performance results for our models under different conditions. Our results suggest that RoBERTa has the edge over others across all evaluated tasks.

## C.4  Aspect-based Sentiment Analysis Benchmark

Our dataset can be naturally used as an aspect-based sentiment analysis (ABSA) benchmark. For each sentence, it may contain up to 4 aspects with respect to the reviewing restaurant. As ABSA benchmarks are usually small and sparse with missing labels, our dataset provides validated aspect-based labels, and is one of the largest human validated ABSA benchmark.

To evaluate model performance, we adapt standard finetuning approach for ABSA benchmarks as proposed by [49]. Instead of single sentence classification, we add another auxiliary sentence representing the aspect. For instance, to predict the label for the 'food' aspect for "the food here is good but not the service", we append a single aspect token with a separator, and construct our input sentence as "the food here is good but not the service [SEP] food". Table 8 shows the performance results for our models under different conditions.

---

[7]We implemented T5 (t5-base; [39]) as a text-to-text model with the goal of treating predicted tokens as class labels. However, this raised unanticipated implementation questions concerning how to post-process multi-token class labels (e.g., "very positive") for use in our explainer methods. As a result, we have elected to leave the T5 results out of the current draft, but we intend to include them in the next version once they have been more thoroughly vetted.

Table 9: Model performance on top-k author identity prediction with number of train and dev examples.

| Model | Accuracy | Macro-F1 | # train | # dev |
|-------|----------|----------|---------|-------|
| Random (k=5) | 0.16 | 0.15 | 1105 | 227 |
| Random (k=10) | 0.10 | 0.10 | 2072 | 519 |
| Random (k=15) | 0.07 | 0.07 | 2963 | 741 |
| RoBERTa (k=5) | 0.27 | 0.16 | 1105 | 227 |
| RoBERTa (k=10) | 0.14 | 0.05 | 2072 | 519 |
| RoBERTa (k=15) | 0.11 | 0.04 | 2963 | 741 |

## C.5 Author Identity Prediction

One potential artifact of our benchmark is edited sentence may expose author identity, which may result in artifact in interpreting model performance. To quantify this potential artifact, we train models to predict author identities based on the sentences. We create author identity prediction dataset by aggregating our dataset by anonymized worker ids. We then split the dataset into train/dev with a 4-to-1 ratio. For model training, we finetune RoBERTa for 5 epochs with a batch size of 32, a learning rate of $2e^{-5}$, and a maximum sequence length of 128. Note that we only consider top-k annotators ranked by their contributions (i.e., number of examples in our dataset). Table 9 shows the performance results of our finetuned models with a random classifier. Our results suggest that potential artifacts may exist but only for a limited extend.

# D Additional Results

In this section, we report additional results for `bert-base-uncased`, `roberta-base`, `gpt-2`, and an `LSTM`, fine-tuned on binary, ternary and 5-way versions of the sentiment task. These models are described in Appendix C. Table 10 summarizes all the results.

We refer to the results section in the main text for an explanation of the different metrics considered. Which metric is best depends on the final use-case and whether it is more important to estimate the direction or the magnitude of the effect.

**ICaCE-cosine** Figure 10 shows the results for the ICaCE-Error with the *cosine* distance metric. The explanation methods that take the direction of the intervention into account (Approx, CONEXP, S-Learner) are the clear winners across all different models considered. S-Learner marginally wins across the most settings, but the conceptually simple Approx baseline is a close second. The strong performance of this simple baseline across the board suggests that most methods perform subpar, and that there is potential value in developing better concept-based model explanation methods.

Both TCAV and ConceptSHAP struggle to achieve better-than-random performance across all settings. Further analysis is needed to exactly understand why these methods are struggling.

Some additional trends emerge that require more analysis to fully understand. For example, Approx generally increases in performance when evaluated on more fine-grained classification settings, while CONEXP is typically worse here.

**ICaCE-normdiff** Figure 11 shows the results for the ICaCE-Error with the *normdiff* distance metric. In general, it is more difficult for explanation methods to estimate the magnitude of the intervention effect when the task increases in complexity. For a given explanation method and model, best results are often achieved for the binary classification problem.

The conceptually simple Approx baseline wins across the board. S-Learner is only able to match its performance a few times. While previous results already showed that most of the methods fall behind the Approx baseline, the results are particularly striking for this metric.

While S-learner and CONEXP were somewhat comparable on the *cosine* metric, their differences become clear on the `normdiff` metric: S-Learner is better at estimating the magnitude of the intervention.

An interesting trend can be observed for TCAV, which has good performance on the binary task but becomes worse than random when evaluated on the ternary and 5-way settings. ConceptSHAP is the only method that consistently breaks the upward trend when going from ternary to the 5-way setting. More analysis is needed to understand both these phenomena.

**ICaCE-L2**    Figure 12 shows the results for the ICaCE-Error with the *L2* distance metric. Because this metric takes both the scale and direction of the effect into account, it is slightly harder to interpret. In general, the performance drops when evaluated on more fine-grained classification settings.

Again, the Approx baseline is a strong contestant, but on this metric the results are more varied. S-Learner is consistently the best at producing the closest explanation in Euclidian distance to the real effect for the 5-way setting.

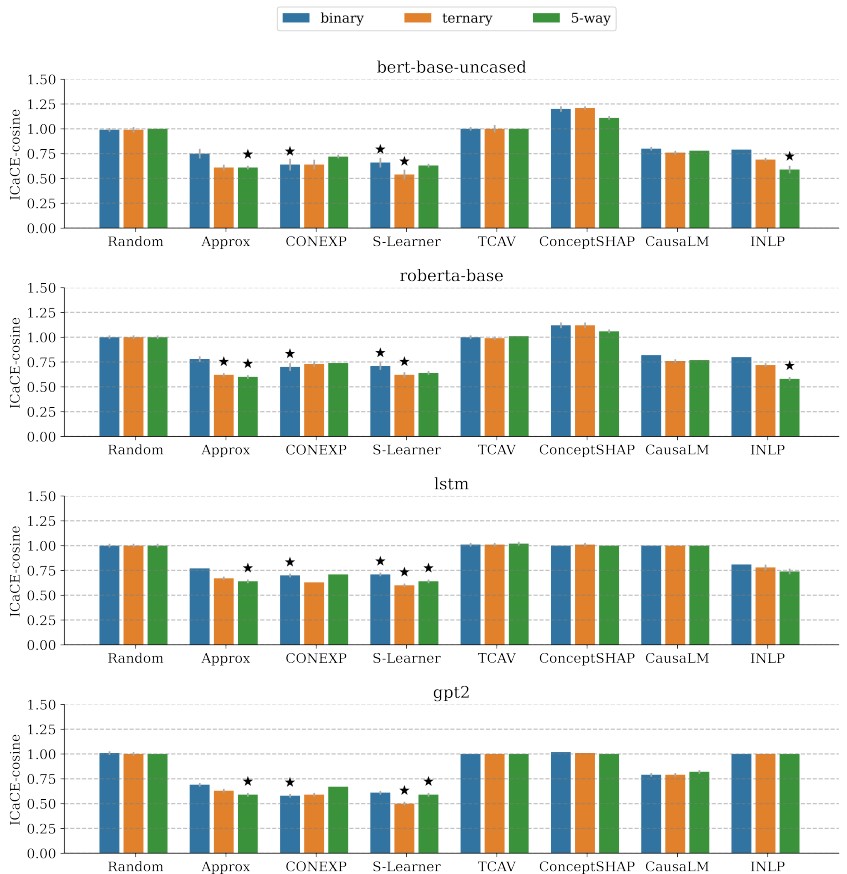

Figure 10: ICaCE-Error for all experiments using the *cosine* distance metric. **Lower is better**. Results averaged over 5 distinct seeds. Error bars (in gray) display the standard deviation. Stars denote the best results for a given classification setting.

# E    CausaLM

## E.1    Our adaptation

The CausaLM algorithm was originally designed to estimate the average treatment effect of a high-level concept on pre-trained language models. Its output estimator is the textual representation averaged treatment effect (TReATE), which is computed as:

$$\text{TReATE}_{\mathcal{N}_\phi}(C; \mathcal{D}) = \frac{1}{|\mathcal{D}|} \sum_{x \in \mathcal{D}} \mathcal{N}'\big(\phi_C^{\text{CF}}(x)\big) - \mathcal{N}\big(\phi(x)\big), \tag{21}$$

Table 10: ICaCE scores on the test set for the binary, ternary and 5-way classification settings. **Lower is better**. Results averaged over 5 distinct seeds; standard deviations in parentheses.

(a) ICaCE scores for 5-way sentiment classification setting.

| Model | Metric | Random | Approx | CONEXP | S-Learner | TCAV | ConceptSHAP | CausaLM | INLP |
|---|---|---|---|---|---|---|---|---|---|
| BERT | $L2_{ICaCE}$ | 0.94 (.01) | 0.81 (.01) | 0.82 (.02) | 0.74 (.02) | 0.82 (.01) | 1.25 (.01) | 0.86 (.01) | 0.80 (.02) |
| | $COS_{ICaCE}$ | 1.00 (.00) | 0.61 (.01) | 0.72 (.01) | 0.63 (.01) | 1.00 (.00) | 1.11 (.01) | 0.78 (.00) | 0.59 (.03) |
| | $NormDiff_{ICaCE}$ | 0.67 (.02) | 0.44 (.01) | 0.62 (.02) | 0.54 (.02) | 0.78 (.02) | 0.56 (.02) | 0.68 (.02) | 0.73 (.02) |
| RoBERTa | $L2_{ICaCE}$ | 0.97 (.01) | 0.83 (.01) | 0.86 (.01) | 0.78 (.01) | 0.85 (.01) | 1.24 (.01) | 0.90 (.01) | 0.84 (.01) |
| | $COS_{ICaCE}$ | 1.00 (.01) | 0.60 (.01) | 0.74 (.00) | 0.64 (.01) | 1.01 (.00) | 1.06 (.01) | 0.77 (.00) | 0.58 (.01) |
| | $NormDiff_{ICaCE}$ | 0.72 (.01) | 0.45 (.01) | 0.67 (.01) | 0.59 (.01) | 0.83 (.01) | 0.61 (.01) | 0.74 (.01) | 0.81 (.01) |
| GPT-2 | $L2_{ICaCE}$ | 0.81 (.02) | 0.72 (.02) | 0.68 (.02) | 0.60 (.02) | 0.68 (.02) | 1.03 (.02) | 0.76 (.02) | 0.72 (.01) |
| | $COS_{ICaCE}$ | 1.00 (.00) | 0.59 (.01) | 0.67 (.00) | 0.59 (.01) | 1.00 (.00) | 1.00 (.00) | 0.82 (.01) | 1.00 (.00) |
| | $NormDiff_{ICaCE}$ | 0.52 (.02) | 0.41 (.01) | 0.47 (.02) | 0.40 (.01) | 0.65 (.02) | 0.46 (.01) | 0.52 (.02) | 0.58 (.03) |
| LSTM | $L2_{ICaCE}$ | 0.89 (.01) | 0.86 (.01) | 0.79 (.01) | 0.73 (.01) | 0.78 (.02) | 1.27 (.04) | 0.76 (.01) | 0.79 (.01) |
| | $COS_{ICaCE}$ | 1.00 (.01) | 0.64 (.01) | 0.71 (.00) | 0.64 (.01) | 1.02 (.01) | 1.00 (.00) | 1.00 (.00) | 0.74 (.02) |
| | $NormDiff_{ICaCE}$ | 0.62 (.01) | 0.50 (.01) | 0.59 (.01) | 0.53 (.01) | 0.70 (.01) | 0.54 (.00) | 0.76 (.01) | 0.60 (.01) |

(b) ICaCE scores for ternary sentiment classification setting.

| Model | Metric | Random | Approx | CONEXP | S-Learner | TCAV | ConceptSHAP | CausaLM | INLP |
|---|---|---|---|---|---|---|---|---|---|
| BERT | $L2_{ICaCE}$ | 0.79 (.01) | 0.54 (.01) | 0.65 (.00) | 0.56 (.00) | 0.56 (.00) | 0.94 (.01) | 0.72 (.00) | 0.58 (.01) |
| | $COS_{ICaCE}$ | 0.99 (.02) | 0.61 (.02) | 0.64 (.04) | 0.54 (.04) | 1.00 (.03) | 1.21 (.01) | 0.76 (.01) | 0.69 (.01) |
| | $NormDiff_{ICaCE}$ | 0.60 (.00) | 0.42 (.01) | 0.54 (.00) | 0.48 (.00) | 0.55 (.00) | 0.62 (.01) | 0.62 (.00) | 0.55 (.01) |
| RoBERTa | $L2_{ICaCE}$ | 0.79 (.01) | 0.56 (.00) | 0.65 (.01) | 0.57 (.01) | 0.55 (.01) | 0.88 (.02) | 0.74 (.01) | 0.55 (.01) |
| | $COS_{ICaCE}$ | 1.00 (.01) | 0.62 (.01) | 0.73 (.02) | 0.62 (.02) | 0.99 (.01) | 1.12 (.01) | 0.76 (.01) | 0.72 (.01) |
| | $NormDiff_{ICaCE}$ | 0.61 (.01) | 0.43 (.00) | 0.54 (.00) | 0.48 (.00) | 0.54 (.00) | 0.61 (.01) | 0.66 (.01) | 0.54 (.01) |
| GPT-2 | $L2_{ICaCE}$ | 0.75 (.01) | 0.57 (.01) | 0.60 (.01) | 0.52 (.01) | 0.52 (.01) | 0.69 (.01) | 0.68 (.01) | 0.61 (.03) |
| | $COS_{ICaCE}$ | 1.00 (.01) | 0.63 (.01) | 0.59 (.01) | 0.50 (.01) | 1.00 (.00) | 1.01 (.00) | 0.79 (.01) | 1.00 (.00) |
| | $NormDiff_{ICaCE}$ | 0.54 (.01) | 0.42 (.01) | 0.47 (.01) | 0.42 (.01) | 0.51 (.01) | 0.52 (.01) | 0.55 (.01) | 0.51 (.01) |
| LSTM | $L2_{ICaCE}$ | 0.76 (.00) | 0.58 (.01) | 0.63 (.01) | 0.55 (.01) | 0.55 (.01) | 1.03 (.04) | 0.53 (.01) | 0.68 (.01) |
| | $COS_{ICaCE}$ | 1.00 (.01) | 0.67 (.01) | 0.63 (.00) | 0.60 (.01) | 1.01 (.01) | 1.01 (.01) | 1.00 (.00) | 0.78 (.02) |
| | $NormDiff_{ICaCE}$ | 0.56 (.01) | 0.45 (.01) | 0.51 (.00) | 0.46 (.01) | 0.51 (.01) | 0.65 (.01) | 0.52 (.01) | 0.56 (.01) |

(c) ICaCE scores for binary sentiment classification setting.

| Model | Metric | Random | Approx | CONEXP | S-Learner | TCAV | ConceptSHAP | CausaLM | INLP |
|---|---|---|---|---|---|---|---|---|---|
| BERT | $L2_{ICaCE}$ | 0.60 (.01) | 0.19 (.01) | 0.51 (.00) | 0.31 (.00) | 0.31 (.01) | 0.76 (.06) | 0.57 (.01) | 0.51 (.05) |
| | $COS_{ICaCE}$ | 0.99 (.01) | 0.75 (.04) | 0.64 (.05) | 0.66 (.04) | 1.00 (.01) | 1.20 (.02) | 0.80 (.01) | 0.79 (.00) |
| | $NormDiff_{ICaCE}$ | 0.52 (.01) | 0.19 (.01) | 0.50 (.01) | 0.30 (.01) | 0.30 (.01) | 0.55 (.05) | 0.56 (.01) | 0.50 (.04) |
| RoBERTa | $L2_{ICaCE}$ | 0.59 (.01) | 0.18 (.01) | 0.51 (.00) | 0.31 (.00) | 0.29 (.01) | 0.68 (.06) | 0.61 (.00) | 0.31 (.01) |
| | $COS_{ICaCE}$ | 1.00 (.01) | 0.78 (.02) | 0.70 (.03) | 0.71 (.03) | 1.00 (.01) | 1.12 (.02) | 0.82 (.00) | 0.80 (.00) |
| | $NormDiff_{ICaCE}$ | 0.52 (.00) | 0.18 (.00) | 0.51 (.00) | 0.31 (.00) | 0.29 (.01) | 0.54 (.04) | 0.60 (.00) | 0.31 (.01) |
| GPT-2 | $L2_{ICaCE}$ | 0.59 (.00) | 0.19 (.01) | 0.50 (.00) | 0.31 (.00) | 0.29 (.00) | 0.39 (.01) | 0.55 (.01) | 0.45 (.01) |
| | $COS_{ICaCE}$ | 1.01 (.01) | 0.69 (.01) | 0.58 (.01) | 0.61 (.01) | 1.00 (.00) | 1.02 (.00) | 0.79 (.01) | 1.00 (.00) |
| | $NormDiff_{ICaCE}$ | 0.51 (.01) | 0.19 (.01) | 0.50 (.00) | 0.31 (.00) | 0.29 (.00) | 0.35 (.01) | 0.53 (.01) | 0.41 (.01) |
| LSTM | $L2_{ICaCE}$ | 0.58 (.01) | 0.20 (.01) | 0.51 (.00) | 0.32 (.01) | 0.31 (.00) | 0.78 (.05) | 0.28 (.00) | 0.47 (.01) |
| | $COS_{ICaCE}$ | 1.00 (.01) | 0.77 (.00) | 0.70 (.01) | 0.71 (.01) | 1.01 (.01) | 1.00 (.00) | 1.00 (.00) | 0.81 (.00) |
| | $NormDiff_{ICaCE}$ | 0.50 (.01) | 0.20 (.01) | 0.50 (.00) | 0.32 (.01) | 0.29 (.00) | 0.64 (.04) | 0.28 (.00) | 0.46 (.01) |

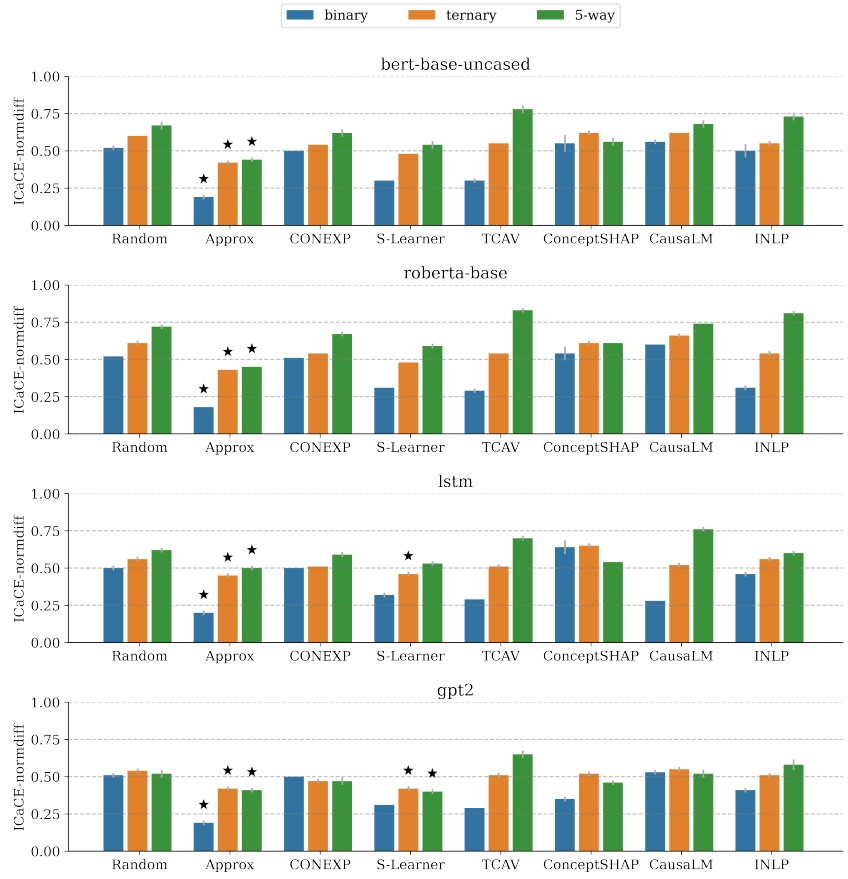

Figure 11: ICaCE-Error for all experiments using the *normdiff* distance metric. **Lower is better**. Results averaged over 5 distinct seeds. Error bars (in gray) display the standard deviation. Stars denote the best results for a given classification setting.

where $\phi_C^{\text{CF}}$ denotes the learned counterfactual representation that information about concept $C$ is not present, $\mathcal{N}'$ is a classifier trained on this counterfactual representation, and $\mathcal{D}$ is a dataset.

However, for comparison on the CEBaB data, we require the estimation of individual causal concept effects (ICaCE). To allow a fair comparison, we swap the TReATE output estimator with TReITE (Equation 10). The only difference between these estimators is that in TReITE we remove the average across $\mathcal{D}$, and output the estimated effect of individual examples.

### E.2 Implementation details

For all counterfactual models, we optimize using the Adam optimizer with `lr=2e-5`, `epochs=3`, `batch_size=48`, and the relative weight of the adversarial task, $\lambda$, is set to $0.1$.

For both the factual models and fine-tuning phase, we optimize using the Adam optimizer with `lr=1e-3`, `epochs=50`, and `batch_size=256`. The differences in hyperparameter values is due to the different architectures we employ; for the counterfactual models we train the entire language model ($\phi$), and for the factual models and the fine-tuning phase we freeze the embedding weights ($\phi$) and train only the classification head ($\mathcal{N}$).

All CausaLM models were trained using 2 Nvidia GTX 1080 Ti 12GB GPUs.

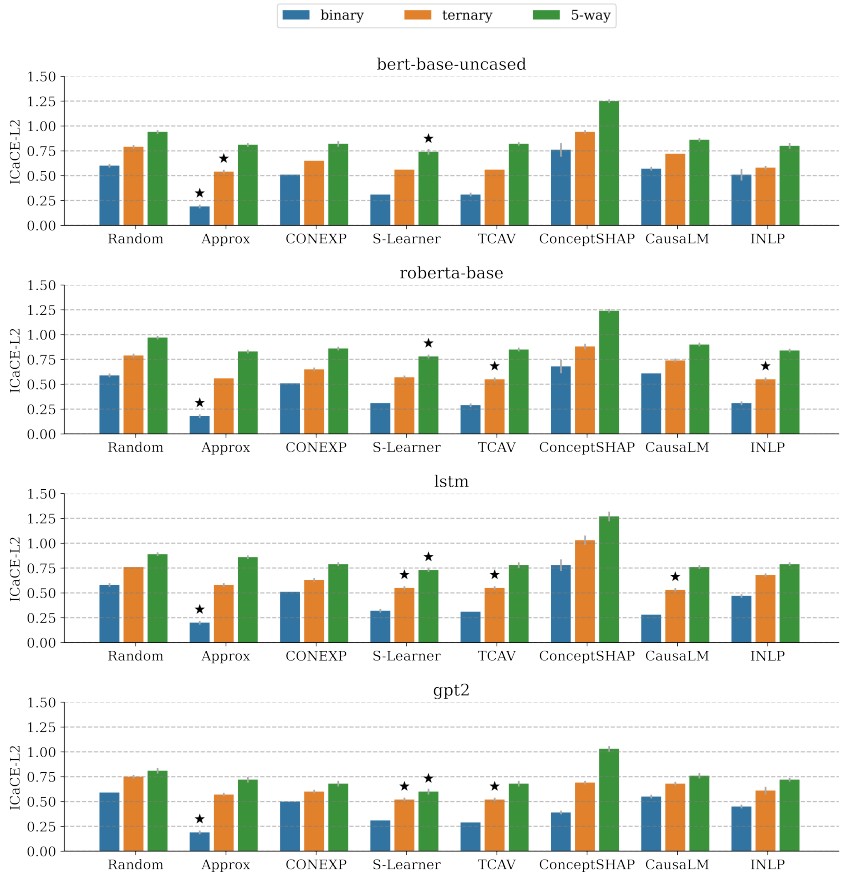

Figure 12: ICaCE-Error for all experiments using the *L2* distance metric. **Lower is better**. Results averaged over 5 distinct seeds. Error bars (in gray) display the standard deviation. Stars denote the best results for a given classification setting.

# F    INLP

## F.1    Our adaptation

The INLP algorithm was originally designed to debias word embeddings by iteratively projecting them onto the null-space of some protected attribute (concept). However, INLP may serve as an estimation method similar to CausaLM, with the two following crucial differences. First, its lack of ability to control for potential confounders. Second, it operates on the representation rather than on the actual model weights. Since CausaLM and INLP share common characteristics, their output estimators are computed in the same way. See §E for extended details.

## F.2    Implementation details

In order to guard for a "protected attribute" (concept), INLP determines whether this concept is present in an embedding or not by learning a linear separator in the embedding space. Following the practice suggested in the original paper, we choose our linear separator to be an SVM learned using SGD with $\alpha = 0.01$, $\varepsilon = 0.001$, and `max_iter=1000`. Logistic regression showed similar behavior. We project the representation to the null-space with respect to the concept 10 times. In fact, and similarly to the original paper, we converge to random accuracy of predicting the concept from the counterfactual representation after 4-5 iterations.

For all concepts, the classification head on top of the language model that trained to predict the overall sentiment labels trains for 5 epochs using the Adam optimizer with `lr=2e-5`.

## G TCAV

### G.1 Our adaptation

The Testing with Concept Activation Vectors (TCAV) explanation method was originally designed to count the percentage of test inputs from dataset $\mathcal{D}$ that are positively influenced by some high-level concept. It outputs a count over the number of examples that are change towards the direction of concept $C$, and computed as:

$$\text{TCAV}_{\mathcal{N}_\phi}(k, C; \mathcal{D}) = \frac{|\{x \in \mathcal{D} : \nabla \mathcal{N}_k(\phi(x)) \cdot v_C > 0\}|}{|\mathcal{D}|}, \tag{22}$$

where $k$ is some class index and $v_C$ is a linear direction in the activation space, given by the coefficients of a linear separator trained to distinguish between examples that include or exclude the concept $C$.

While TCAV's output is a count over examples, we use the raw sensitivity (directional derivative). This approach is supported by the authors of the original paper: "one could also use a different metric that considers the magnitude of the conceptual sensitivities" [26]. Also, since TCAV operates on the gradients of a model's logits but the ICaCEs are the difference of two probability vectors, we normalize its outputs by taking Tanh.

### G.2 Implementation details

To learn the Concept Activation Vector (CAV, i.e., a linear direction in the activation space of $\phi$), we train a linear separator to distinguish between examples that include the concept (labeled positive or negative) and examples that do not include it (labeled unknown). When learning CAVs, we drop all CEBaB train examples that are not labeled for aspect (concept) or do not have a majority with respect to the aspect.

Identically to the original paper, our CAV linear separator is an SVM learned using SGD with $\alpha = 0.01, \varepsilon = 0.001$ and max_iter $= 1000$.

## H ConceptSHAP

### H.1 Our adaptation

The original ConceptSHAP algorithm takes a complete set of concepts $C \in \{C_1, ..., C_m\}$ (such that its completeness score in Equation 25 is higher than some threshold) and outputs the relative contribution to the test accuracy of each $C_i$. It outputs an estimator given by the following formula

$$\text{Shapley}_{\{C_1,...,C_m\}}(C) = \sum_{S \subseteq \{C_1,...,C_m\} \setminus C} \frac{(m - |S| - 1)! \, |S|!}{m!} [\eta(S \cup \{C\}) - \eta(S)], \tag{23}$$

where $\eta$ is a scoring function operating on sets of concepts that output accuracy ratios.

Similarly to the other methods, if $\eta$ outputs accuracy ratios, then the output of ConceptSHAP is not a suitable estimator for ICaCE. Our straightforward adaptation for ConceptSHAP is to make $\eta$ output class probabilities for classes instead of accuracy ratios.

Our adapted version outputs a vector for each $C \in \{C_1, \ldots, C_m\}$ and $x$ according to the following equation:

$$\text{ConceptSHAP}_{\mathcal{N}_\phi}(C; x) = \sum_{S \subseteq \{C_1,...,C_m\} \setminus C} \frac{(m - |S| - 1)! \, |S|!}{m!} [\eta(S \cup \{C\}) - \eta(S)], \tag{24}$$

where $\eta$ is a function defined as $\eta_{\mathcal{N}_\phi}(S) = \sup_g \mathcal{N}\big(g\big(V_S \, \phi(x)\big)\big)$, and $V_S$ is a matrix with the learned concept directions as its rows $V_S = \big(v_C^T\big)_{C \in S} \in \mathbb{R}^{|S| \times h}$.

Yeh et al. [57] calculate concept directions $v_{C_j}$ automatically by learning a neural network classifier. To allow for a fair comparison between ConceptSHAP and the other evaluated methods, we use the

concept activation vectors $v_{C_1}, \ldots, v_{C_m}$ as the input concepts (similarly to those used in Kim et al. [26]).

In addition, in the original paper the authors learn the concepts $v_C$ automatically, by using a carefully constructed loss function. To allow a fair comparison, we learn the concept vector by exploiting our labeled aspects (concepts), in a way similar to TCAV. See Section G.2 for more details.

## H.2  Completeness Scores of Treatment Concepts

Given a feature representation $\phi$ and a classification head $\mathcal{N}$, the completeness score is defined by:

$$\text{completeness}_{\mathcal{N}_\phi}(S; D, Y) = \frac{\sup_g \frac{1}{|\mathcal{D}|} \sum_{(x,y) \in \mathcal{D}, Y} \mathbb{1} \left[ y = \arg \max_{y'} \mathcal{N}_{y'} \left( g \left( V_S \, \phi(x) \right) \right) \right] - a_r}{\frac{1}{|\mathcal{D}|} \sum_{(x,y) \in \mathcal{D}, Y} \mathbb{1} \left[ y = \arg \max_{y'} \mathcal{N}_{y'} \left( \phi(x) \right) \right] - a_r},$$

(25)

where $a_r$ is is the accuracy of a classifier that outputs random predictions, $S \subseteq \{C_1, ..., C_m\}$ and $V_S$ is a matrix with the learned concept directions as its rows $V_S = \left( v_C^T \right)_{C \in S} \in \mathbb{R}^{|S| \times h}$.

For all models, the completeness we get for the set of concepts $S = \{\text{ambiance, food, service, noise}\}$ is larger than $0.9$.

## H.3  Hyperparameters

The hyperparameters for CAV are identical to those of TCAV (Section G.2). To calculate $\eta$ and the completeness score, we follow the original paper and set $g$ to be a two-layer perceptron with 500 hidden units, learned using Adam optimizer for 50 epochs, employing `lr=1e-2` and `batch_size=128`.