# OpenReview forum: "CEBaB: Estimating the Causal Effects of Real-World Concepts on NLP Model Behavior"
_NeurIPS.cc/2022/Conference — NeurIPS 2022 Accept_

### Official Review · Reviewer_NcbX · 2022-07-11

**Rating:** 7
**Confidence:** 4
**Soundness:** 3 good
**Presentation:** 2 fair
**Contribution:** 2 fair

**Summary:**

This paper introduces CEBaB, a new benchmark dataset to assess model explanations for NLP models. The key idea here is to edit short restaurant reviews with human generated counterfactual reviews in which one aspect of the review is modified. With aspect level and review level sentiment annotations on CEBaB, authors are able to approximate how a particular model might weigh some abstract concepts more than others when performing sentiment classification. The authors evaluate six (existing and proposed) explanation methods on this benchmark and find that there are significant differences between them. I like that this is a simple and intuitive idea that appears to work well in practice. However, there is a disconnect between the general claims the paper makes and the benchmark. The claims are about NLP model behavior in general but should be constrained to sentiment analysis. I have updated my sore following author response.

**Questions:**

- I don't understand what Table 4 is. The caption is very confusing so if you could share that, that'll be great.

**Limitations:**

I would like a greater discussion of how the method could be extended to other more complicated tasks. And if there are challenges associated with that, what are those?

**Strengths And Weaknesses:**

### Strengths:
- This paper addresses an important problem of evaluating explainability methods. The paper is well motivated and well written.
- The conceptual framing offered by this paper as well as the resource contribution could be useful to future researchers.
- The authors have also provided an extensive experimental analysis of how well multiple model explanation methods are able to capture the true concept effect.
- I like the focus on direction of an intervention's effect on model behavior rather than the scale of effects as it simplifies the problem to knowing what causes what rather than the scale of a cause's contribution to the effect.


Weaknesses:
- This work could be much better positioned in prior work on benchmarks for explainability and counterfactual analysis for model explanations (such as the Language Interpretability Toolkit).
- The benchmark is only for sentiment analysis which limits its use while broader claims are made about NLP explainability. While that's still a good first step, it is arguably easier to "explain" a model on a sentiment analysis task versus say NLI or other more complicated NLP tasks using CEBaB, and there is no discussion offered in the paper on how this would extend to those tasks.

---

> ### Author Response · Authors · 2022-08-02
> **Official Response to Reviewer NcbX (Part 1/2)**
>
> We greatly appreciate the reviewer’s in-depth feedback on our paper, which encouraged us to consider the way we are characterizing our goals and findings.  Below we provide a point-to-point response, which adds to our above response to the shared reviewer themes and the revised version of the paper submitted alongside our response.
>
> ***
>
> > **Q1**: “This work could be much better positioned in prior work on benchmarks for explainability and counterfactual analysis for model explanations (such as the Language Interpretability Toolkit).”
>
> **A1**: Agreed. In addition to the revisions we made to **Section 2 - Prior Work** following this comment, we would like to further elaborate on the differences between our work and other predecessors.
>
> - [11, CausalLM] estimates the causal effect of high-level concepts on the model, using synthetic data, by considering the aggregated CaCE values. One crucial advantage of CEBaB is that it evaluates explanation methods with naturalistic data rather than synthetic data. This allows us to compare different explanation methods against a human-validated ground-truth. Additionally, we define our final score on the level of individual examples (ICaCE), where CausaLM defines scores at the aggregated CaCE level. This allows us to better study nuanced local explanations of the model behavior, compared to aggregated claims. Furthermore, we consider a range of different metrics (cosine, normdiff, L2), which allows us to e.g. detangle the explanation method's ability to estimate directions of changes and magnitudes of changes.
>
> - [52, Language Interpretability Toolkit (LIT)] is a new reference added per a suggestion by R4. LIT is “an open-source platform for visualization and understanding of NLP models”. It is an explanation toolkit rather than an explanation benchmark. The closest feature to causal concept-based explanations provided by LIT is a family of counterfactual analyses for NLP models. While LIT only provides rule-based tools to generate counterfactuals from an existing dataset (e.g., “HotFlip”), our benchmark contains human-written counterfactuals.  Lastly, while LIT only allows for a qualitative analysis, CEBaB offers quantitative analyses in the form of ICaCE scores using different metrics.
>
> - [57 (formerly 56), ConceptSHAP] provides mostly qualitative comparisons by asking human judges to score different concepts that the explanation method discovered. In our work, we propose to use the ICaCE score which measures concept-based explanation quantitatively by calculating the distance between the true and estimated causal effects. Although [57, ConceptSHAP] provides a quantitative evaluation as well, it uses a synthetic image dataset without rigorous control of the dataset generation process, and uses only non-causal, accuracy-like metrics to benchmark the explanation methods.
>
> The above three points are reflected in the revised paper in the following quote: *“Other works that do compare to some ground-truth either employ a non-causal evaluation scheme [26], use causal evaluation metrics which do not capture performance on individual examples [52], evaluate on synthetic counterfactuals and rule-based augmentations [11 , 52], or are tailored for a specific explanation method and are hard to generalize [57].”*

---

> > ### Comment · Reviewer_NcbX · 2022-08-07
> > **Response to your response.**
> >
> > "We find that explanation performance on CEBaB is already lackluster, indicating that this will also be the case for more complicated tasks. Thus, we believe that CEBaB will serve as a call to action for the NLP model explainability community and that our resources (data and methodology) will help the development and validation of new explanation techniques, which will subsequently have an impact outside the scope of English sentiment analysis."
> >
> > I don't see how or why CEBaB will serve as a call to action for the NLP model explainability community given most of the current methods are meaningless and there's no reason to expect them to be explaining anything absent unrealistic assumptions. That said, the dataset is definitely a positive step towards analyzing these methods. Finally, you point to a case study in the response as to establish how this methodology could be extended but I don't see it anywhere in the paper. I'm inclined to maintain my score as is.

---

> > > ### Author Response · Authors · 2022-08-07
> > > **CEBaB can spur the next generation of more robust explanation methods**
> > >
> > > Thanks for the comments! You've raised great points that we feel we can be responsive to:
> > >
> > > 1. We think that having a benchmark with dense, human-created counterfactual texts with lots of labels is just what is needed to get the field out of its current rut. To date, people developing explanation methods have had to rely on general intuition, synthetic benchmarks, and custom tasks that don't support apples-to-apples comparisons. With CEBaB, methods can be evaluated with fixed criteria. In our experience, if you build high-quality benchmarks, they attract people to work on the area, and we predict CEBaB can be such a nudge for people. Conversely, it is not surprising that causal explainer work has been somewhat at sea to date given the lack of clear evaluation criteria before CEBaB.
> > >
> > > 2. The assumptions behind CEBaB seem reasonable to us. The aspect-level categories are originally from OpenTable. It's unusual to have our density of counterfactual texts with labels, but that's the special value of the benchmark -- that it provides a measurement tool that would not normally be available in the wild.
> > >
> > > 3. In terms of the general framework: we sketched out an extension to spam/ham and would be happy to go further. We are still thinking about where this would fit into the paper. We've released all our MTurk templates, and minor tweaks to those will support a wide range of classification tasks. One just needs to think creatively about the causal graph that one wants to be able to explore.

---

> > > > ### Comment · Reviewer_NcbX · 2022-08-07
> > > > **Thanks for the response**
> > > >
> > > > Do you anticipate potential scenarios where a method works well on this benchmark but is not actually better generally (say on Question Answering or NLI or hate speech detection)?

---

> > > > > ### Author Response · Authors · 2022-08-07
> > > > > **Can estimators control for confounds?**
> > > > >
> > > > > Also an interesting conceptual question. The scenario we've thought about the most concerns controlling for confounds. Some methods are explicitly motivated by their ability to do this. The default exclusive train set for CEBaB might not have rich enough confounds to bring this out. So someone advocating for a confound-controlling method might feel it is misleading if they tie with a method that cannot do this.
> > > > >
> > > > > However, CEBaB's inclusive train set is so rich in counterfactual texts that one can create datasets with lots of confounds in them, just by sampling using the labels and other metadata. We thought about constructing such confounding datasets for the paper, but we ran out of room. However, if people want to create such versions of CEBaB, that seems great, especially if they release the split information so that others can compare.
> > > > >
> > > > > The above isn't quite a scenario where a simple method does _better_ but isn't better in general. If a sophisticated method can handle confounded datasets but not simple scenarios, it seems like an ambiguous win for that method, especially since we won't know how hard our problems are in general or how confounded they are in general.
> > > > >
> > > > > Ideally, CEBaB will end up being used to test many scenarios and the best methods will be able to handle all of them.

---

> ### Author Response · Authors · 2022-08-02
> **Official Response to Reviewer NcbX (Part 2/2)**
>
> > **Q2**: “The benchmark is only for sentiment analysis which limits its use while broader claims are made about NLP explainability. While that's still a good first step, it is arguably easier to "explain" a model on a sentiment analysis task versus say NLI or other more complicated NLP tasks using CEBaB, and there is no discussion offered in the paper on how this would extend to those tasks. I would like a greater discussion of how the method could be extended to other more complicated tasks. And if there are challenges associated with that, what are those?”
>
> **A2**: While CEBaB focuses on English sentiment analysis, a relatively easy task, its potential impact as a human-validated natural model explainability benchmark is not limited to this task. The reviewer makes the observation that some tasks are arguably more difficult to explain. We find that explanation performance on CEBaB is already lackluster, indicating that this will also be the case for more complicated tasks. Thus, we believe that CEBaB will serve as a call to action for the NLP model explainability community and that our resources (data and methodology) will help the development and validation of new explanation techniques, which will subsequently have an impact outside the scope of English sentiment analysis.
>
> Once satisfactory results are achieved on CEBaB, we hope the community will use our resources as a starting point towards considering more complicated tasks, as the reviewer suggested. Our evaluation methodology and proposed metrics are directly applicable to other tasks such as NLI. The main challenge will be sourcing a new set of human-generated counterfactuals for the specific task, where deciding which concepts to annotate is an important task-specific decision.
>
> Lastly, to respond to the question about extensibility, we also provide a case study where we can build a CEBaB-version of the classic spam classification problem in ML:
>
> - **Task**: Spam or Ham Classification. The input to this task is an email with a label indicating whether this email is a spam or not a spam.
>
> - **Concepts**: The email contains rich high-level concepts, which can be divided into two categories: **(1) linguistic concepts**: grammaticality, fluency, vocabulary richness, punctuation usage, etc. **(2) semantic concepts**: sentiment, category, ambiguation, tone, stance, sarcasm, intent, etc. High-level concepts are usually latent to the input (i.e., cannot be expressed in one word or one token directly). Each concept can be associated with a label. For instance, a binary or a discrete variable indicating the grammaticality of the email.
>
> As we can see, it just needs to follow CEBaB’s data collection process and can be created as another concept-based explanation benchmark for spam filtering models.
>
> ***
>
> > **Q3**: “I don't understand what Table 4 is. The caption is very confusing so if you could share that, that'll be great.”
>
> **A3**: We updated the caption of Table 4 in the rebuttal revision, and we hope this clears things up. As we are now focusing on the 5-way problem, we also update Table 4 accordingly in a way that is easy for readers to parse our results. Additionally, below we put CaCE in the context of other metrics, which might also shed more light on what it is.
>
> - **ATE** is a metric that only depends on the data (no model, no explanation method). It measures the causal effect of changing a concept in a given text by measuring the resulting change in the label.
>
> - **CaCE** is a metric that depends on the data and the model (not on the explanation method). It measures how much a concept affects the output of a model. For example, what happens if we change food from positive to negative - how would the output probabilities of the model change? We can think of this as a ground-truth effect that our explainers will try to estimate. The **ICaCE** (Individual CaCE) is the change in model output for a single intervention.
>
> - **ICaCE-error** is a metric that depends on the data, the model, and the explanation method. Given a specific instance of a model, it measures the average distance between the estimated effect for a single intervention (outputted by the explanation method) and the actual effect of that single intervention (the ICaCE).

---

### Official Review · Reviewer_Jcm3 · 2022-07-12

**Rating:** 6
**Confidence:** 4
**Soundness:** 2 fair
**Presentation:** 3 good
**Contribution:** 3 good

**Summary:**

The paper introduced CEBaB, a new  benchmark dataset for assessing concept-based explanation methods in Natural Language Processing (NLP). CEBaB consists of short restaurant reviews with human-generated counterfactual reviews in which an aspect (food, noise, ambiance, service) of the dining experience was modified.

**Questions:**

How this paper help developing evaluation techniques for other problems apart from sentiment analysis?

**Limitations:**

They have described the limitations.

**Strengths And Weaknesses:**

Strengths:
Evaluating explanation methodology for causal effect is important for the research community. The dataset would be useful for that purpose.

Weakness:
The dataset evaluates the model explanation methods based on aspects (food, noise, ambiance, service). This dataset can be used for sentiment analysis. I am not sure if this dataset can help in other classification problems for evaluating causal effects.

---

> ### Author Response · Authors · 2022-08-02
> **Official Response to Reviewer Jcm3**
>
> We greatly appreciate the reviewer’s in-depth feedback on our paper, which encouraged us to consider the way we are characterizing our goals and findings.  Below we provide a point-to-point response, which adds to our above response to the shared reviewer themes and the revised version of the paper submitted alongside our response.
>
> ***
>
> > **Q1**: “The dataset evaluates the model explanation methods based on aspects (food, noise, ambiance, service). This dataset can be used for sentiment analysis. I am not sure if this dataset can help in other classification problems for evaluating causal effects.”
>
> > **Q2**: “How does this paper help developing evaluation techniques for other problems apart from sentiment analysis?”
>
> **A1 & A2**: While the data collected is not directly applicable to tasks other than sentiment analysis, the main data collection process, methodology and conclusions of this work are highly relevant for the general model explainability field. Primarily, we show that existing explanation methods fail our relatively simple task (English sentiment analysis). We hope that this serves as a call to action for the NLP explainability community, which can now leverage our dataset to develop more principled explainability approaches. Additionally, the evaluation and data creation framework proposed in this work are extendible to other tasks and languages. Therefore, we believe that our contributions will generalize beyond the scope of English sentiment analysis.

---

### Official Review · Reviewer_T7tB · 2022-07-17

**Rating:** 6
**Confidence:** 4
**Soundness:** 3 good
**Presentation:** 3 good
**Contribution:** 3 good

**Summary:**

The paper introduces CEBaB, a new benchmark dataset for assessing concept-based explanation methods in the domain of Natural Language Processing (NLP).
CEBaB consists of short restaurant reviews with human-generated counterfactual reviews in which an aspect has been modified.
CEBaB is leveraged to compare various concept-based explanation methods.
The purpose of the paper is to establish natural metrics for comparative assessments of these methods.
More precisely, five leading concept-based explanation methods: CONEXP, TCAV, ConceptSHAP, INLP, and CausaLM.
The dataset of reviews is composed of four aspects (food, service, ambiance, noise), associated with binary evaluation and an overall evaluation estimated from 1 to 5 stars.
The dataset is crowdsourced from an initial set of OpenTable reviews.
As a first step, the crowdsourcers have to expand the reviews to focus on one specific aspect.
In a second step, the crowdsourcers have labeled each aspect of the edited reviews.
The notion of the concept associated with the reviews is not clear, I presume it will be words.
The authors have used it as a classifier to explain a fine-tuned pretrained language model to predict the overall binary sentiment of all restaurant reviews

**Questions:**

Can you please detail the definition of the concept in the dataset? This point is central to all evaluated explanation methods.
Did you consider augmenting the dataset through paraphrasing, do you think it would cause an additional challenge?

**Limitations:**

I do not see a strong limitation in regard to the purpose of the paper and the associated contribution and results.

**Strengths And Weaknesses:**

Strengths.
 * The paper addressed a pertinent question of the explainability of large neural models
 * The paper is clear and properly federate in a unified description method a relatively large spectrum of explanation methods
 * The dataset is large and addressed the problem of explanation in NLP which is particularly known as tedious
 * The dataset has been crowdsourced in two separated steps which is a reasonable indicator of the limited noise in the data.
 * The paper already compares a large set of explanation methods with the resulting dataset

Weakness
 * Unfortunately, the dataset is only in English, as most of the pretrained models are now multi-lingual it could have been interesting to consider this point.
 * The notion of concept is not clear in the context of the reviews, I assume it is the tokens.

---

> ### Author Response · Authors · 2022-08-02
> **Official Response to Reviewer T7tB**
>
> We greatly appreciate the reviewer’s in-depth feedback on our paper, which encouraged us to make significant additions to the paper and better articulate its scope.  Below we provide a point-to-point response, which adds to our above response to the shared reviewer themes and the revised version of the paper submitted alongside our response.
>
> ***
>
> > **Q1**: “Unfortunately, the dataset is only in English, as most of the pretrained models are now multi-lingual it could have been interesting to consider this point.”
>
> **A1**: Extending CEBaB to a multi-lingual setting would be an interesting endeavor. Regardless of the language, we want to highlight that concept-based explanation methods are performing subpar even on the relatively simple task of English sentiment analysis. Before investing resources in extending CEBaB to multiple languages, we hope that the introduction of CEBaB will prompt the model explainability community to develop and validate methods that actually work on our current benchmark. Once the field is ready to consider explanation techniques for different languages and tasks, the data collection and method validation framework provided by this work can be applied there.
>
> ***
>
> > **Q2**: “The notion of concept is not clear in the context of the reviews, I assume it is the tokens. Can you please detail the definition of the concept in the dataset? This point is central to all evaluated explanation methods.”
>
>  **A2**: Concept is not equivalent to tokens. We consider concepts as high-level abstract constituents that are latent in the input but can be extracted[57]. In the context of CEBaB, which consists of restaurant reviews, the concepts are different restaurant aspects (e.g., food, ambiance, service, noise). Each of these concepts can be expressed across multiple tokens and even sentences, so there is no clear 1-to-1 mapping of latent concepts and tokens. Following the reviewer’s comment, we revised the text to better reflect this definition.
>
> ***
>
> > **Q3**: “Did you consider augmenting the dataset through paraphrasing, do you think it would cause an additional challenge?”
>
> **A3**: This is an interesting proposal. In this work, we use counterfactual examples to validate explanation methods. While it would be possible to augment counterfactual examples to build a bigger validation set, this would go against our goal of building a high-quality human-created validation dataset. In follow-up work, where we plan to leverage CEBaB’s interventional data to explicitly train better explanation methods, augmentation strategies could be highly valuable to build more data-efficient explainers.

---

### Official Review · Reviewer_dBGa · 2022-07-17

**Rating:** 6
**Confidence:** 3
**Soundness:** 3 good
**Presentation:** 3 good
**Contribution:** 2 fair

**Summary:**

This paper introduces a new benchmark called CEBaB for evaluating concept-based explanation methods. This benchmark dataset consists of 15,089 restaurant reviews obtained by editing 2,299 original reviews from Opentable. Crowdworkers are asked to modify an original in order to alter a specific aspect (food, ambiance, service, or noise) in the review while holding all other aspects constant. This process creates pairs of examples where only one aspect of them is different, which allows estimating the causal effect of changing the particular aspect. The paper evaluates 6 different explanation methods by measuring the distance of the concept-based explanations w.r.t. the estimated causal effect.


**Questions:**

*Benchmarks:*

Have the authors verified whether edits affect other aspects? The paper verifies that the edits successfully achieve the aspect-level goal (altering a particular aspect towards a specific direction), but there are no descriptions on whether other aspects are constant, which can be done with the help of crowdworkers on a small number of examples in the benchmark

Are the final aggregated review and model score sensitive to different edits of a particular aspect-level goal? For each original review and a specific aspect-level goal, the paper collects one edited text. Ideally, it’s better to have multiple samples (as stated in Eq (2)) for estimating the ICaCE and ATE. It might be worthwhile to collect multiple edits for a relatively small set of examples and investigate the sensitivity.

*Experiments:*

The paper benchmarks the performance of different explanation techniques and presents some interesting findings. E.g. TACV and ConceptShap performs worse (or on par) with the random baseline, even on the straightforward food aspect. It might be worth elaborating or providing qualitative analysis of the failures of these methods.

*Related work*

While the central contribution of this paper is benchmarking. The paper mainly discusses different explanation techniques. It might be good if there could be more discussion on the evaluation side, especially metrics and experimental findings. E.g., whether other work [11,56] uses the same or mostly similar evaluation metrics but evaluates on synthetic tasks.



**Limitations:**

See questions.


**Strengths And Weaknesses:**

Strengths:

The paper collects a natural language dataset with pairs of human-written counterfactuals, which allows building.

The experiments cover an array of popular explanation techniques with varying requirements of access to the models.

The writing is clear and easy to follow.

Weakness:

While the collected benchmark and the experiments of evaluating different concept-based explanation methods are a central contribution, the paper does not provide some important details .

The paper does not provide thorough discussion of results.

The paper does not clearly discuss the relationship with other work on evaluating concept-based explanation methods.

---

> ### Author Response · Authors · 2022-08-02
> **Official Response to Reviewer dBGa (Part 1/2)**
>
> We greatly appreciate the reviewer’s in-depth feedback on our paper, which encouraged us to consider the way we are characterizing our goals and findings.  Below we provide a point-to-point response, which adds to our above response to the shared reviewer themes and the revised version of the paper submitted alongside our response.
> ***
> > **Q1**: “The paper does not clearly discuss the relationship with other work on evaluating concept-based explanation methods. While the central contribution of this paper is benchmarking. The paper mainly discusses different explanation techniques. It might be good if there could be more discussion on the evaluation side, especially metrics and experimental findings. E.g., whether other work [11,56] uses the same or mostly similar evaluation metrics but evaluates on synthetic tasks.”
>
> **A1**: Agreed. In addition to the revisions we made to **Section 2 - Prior Work** following this comment, we would like to further elaborate on the differences between our work and other predecessors.
>
> - [11, CausalLM] estimates the causal effect of high-level concepts on the model, using synthetic data, by considering the aggregated CaCE values. One crucial advantage of CEBaB is that it evaluates explanation methods with naturalistic data rather than synthetic data. This allows us to compare different explanation methods against a human-validated ground-truth. Additionally, we define our final score on the level of individual examples (ICaCE), where CausaLM defines scores at the aggregated CaCE level. This allows us to better study nuanced local explanations of the model behavior, compared to aggregated claims. Furthermore, we consider a range of different metrics (cosine, normdiff, L2), which allows us to e.g. detangle the explanation method's ability to estimate directions of changes and magnitudes of changes.
>
> - [52, Language Interpretability Toolkit (LIT)] is a new reference added per a suggestion by R4. LIT is “an open-source platform for visualization and understanding of NLP models”. It is an explanation toolkit rather than an explanation benchmark. The closest feature to causal concept-based explanations provided by LIT is a family of counterfactual analyses for NLP models. While LIT only provides rule-based tools to generate counterfactuals from an existing dataset (e.g., “HotFlip”), our benchmark contains human-written counterfactuals.  Lastly, while LIT only allows for a qualitative analysis, CEBaB offers quantitative analyses in the form of ICaCE scores using different metrics.
>
> - [57 (formerly 56), ConceptSHAP] provides mostly qualitative comparisons by asking human judges to score different concepts that the explanation method discovered. In our work, we propose to use the ICaCE score which measures concept-based explanation quantitatively by calculating the distance between the true and estimated causal effects. Although [57, ConceptSHAP] provides a quantitative evaluation as well, it uses a synthetic image dataset without rigorous control of the dataset generation process, and uses only non-causal, accuracy-like metrics to benchmark the explanation methods.
>
> The above three points are reflected in the revised paper in the following quote: *“Other works that do compare to some ground-truth either employ a non-causal evaluation scheme [26], use causal evaluation metrics which do not capture performance on individual examples [52], evaluate on synthetic counterfactuals and rule-based augmentations [11 , 52], or are tailored for a specific explanation method and are hard to generalize [57].”*
>
> ***
>
> > **Q2**: “Have the authors verified whether edits affect other aspects? The paper verifies that the edits successfully achieve the aspect-level goal (altering a particular aspect towards a specific direction), but there are no descriptions on whether other aspects are constant, which can be done with the help of crowdworkers on a small number of examples in the benchmark.”
>
> **A2**: This is an excellent suggestion. As a preliminary step toward such an assessment, we sampled 3K training examples and re-validated the concept-level ratings for non-target concepts using our standard validation pipeline using MTurk. We treat each reviewer label as a one-hot vector over the three aspect-level classes and calculate the distance between the centroids of these vectors across our original and re-validated labels. This leads to an average distance of 0.32 (where the minimum is 0 and the maximum is sqrt(3)), suggesting that editing does not disrupt non-target concepts. Indeed, we suspect that this 0.32 score primarily reflects individual differences in sentiment ratings in general and is not especially tied to the editing process. To further explore this, we sampled a separate set of 200 examples and manually evaluated whether editing had affected a non-target concept. From the 200 examples we inspected, we found that 89.5% were unaffected by editing.

---

> ### Author Response · Authors · 2022-08-02
> **Official Response to Reviewer dBGa (Part 2/2)**
>
> > **Q3**: “Are the final aggregated review and model score sensitive to different edits of a particular aspect-level goal? For each original review and a specific aspect-level goal, the paper collects one edited text. Ideally, it’s better to have multiple samples (as stated in Eq (2)) for estimating the ICaCE and ATE. It might be worthwhile to collect multiple edits for a relatively small set of examples and investigate the sensitivity.”
>
> **A3**: This is a valuable observation. We conducted further analyses to address this concern. CEBaB includes 176 examples that have a paired edit (i.e., an extra edit with the same goal and type on the same original sentence, performed by a different worker). The difference in average review score assigned by the workers across these 176 pairs is on average 0.78 stars. This result suggests that most of the paired edits have a high agreement in the final review score, indicating a limited sensitivity. We report this and supplementary analysis in Appendix B8.
>
> ***
>
> > **Q4**: “The paper benchmarks the performance of different explanation techniques and presents some interesting findings. E.g. TACV and ConceptSHAP perform worse (or on par) with the random baseline, even on the straightforward food aspect. It might be worth elaborating or providing qualitative analysis of the failures of these methods.”
>
> **A4**: We agree and plan to provide more qualitative analyses of the failures of these methods in the next revision. To provide more insights into our results, we already updated our main result section and figure to include more results for different versions of our metric, and we now focus on the 5-way problem, which brings out a lot more differences between methods and is overall more robust (as discussed in the shared response). Additionally, we’ve updated Appendix D - Additional Results to include more insights about the methods across models, classification settings, and evaluation metrics.
>
> An initial analysis we plan to consider for the next revision would break down the ICaCE-scores across different aspects and intervention directions. This would allow us to study if some methods are failing on specific aspects or even specific aspect directions, as opposed to the current global aggregation of the ICaCE-scores that only indicates how methods are performing across the board.

---

> ### Comment · Reviewer_dBGa · 2022-08-05
> **Thank you for the clarifications. I'm raising my score.**
>
> Thank you for the clear answers and the added verification experiments. I am raising my score significantly, from 4 to 6.
>
> While I do feel 1) the qualitative analysis to be added could be valuable, and 2) the scope of this dataset is a somewhat limited (sentiment analysis is a somewhat easy task), the new version have explained many missing details and proved more solid evidences of the quality of the dataset, compared to the previous one.

---

### Author Response · Authors · 2022-08-02
**Meta Response to All Reviewers**

We thank the reviewers for their efforts and valuable comments, which have let us make major revisions to the paper, particularly by collecting new crowdsourced data and performing new quantitative analyses. These are already reflected in the rebuttal revision and will continue shaping the way we revise the work over the next weeks.

**We would first like to discuss the shared themes of the reviews**.

First, there seems to be a consensus that the CEBaB dataset is useful for evaluating a large spectrum of explanation methods, under the conceptual framing of the problem as a causal estimation challenge. Like the reviewers, we too believe that a large, human-written counterfactual-based resource could be useful to the DL/ML community, especially in the important area of model explanations, and CEBaB is the only such resource at present as far as we know

Second, while we agree that CEBaB should be extended beyond English sentiment analysis, our results show that current explanation methods struggle even in this relatively simple domain. In fact, we have **a simple baseline method that outperforms all other explanation methods**. This finding suggests that fundamental research still needs to be done before we can trust explanation methods in more complex domains. However, when appropriate, the methodology developed in this paper (both in terms of data collection and evaluation methodology for explainers) can be directly extended to more complex domains and different languages.

Put another way, while there are high-quality algorithms for English sentiment analysis, concept-based explanation in this domain is very much an open problem. In our view, CEBaB is just the first step towards encouraging our community to create interventional benchmarks for the evaluation of concept-based explanation methods.

**Overview of major changes (now appearing in the rebuttal revision)**:

- **Section 2 - Previous Work** now better positions CEBaB within prior work on explainability benchmarks.
- **Section 6 - Experiments and Results.** The reviewers asked for more details on the results and what they mean. While we are still highly space-constrained, we believe we have made major improvements here. The overall takeaways remain the same: explanation methods have a long way to go before they can explain causal effects even in simple domains like CEBaB.
    - We introduced a new baseline that approximates true counterfactuals by sampling from training data. **This baseline is the new best-performing method**. The fact that a simple baseline outperforms popular explanation methods highlights the need for a benchmark like CEBaB. We believe that this finding has **implications beyond English sentiment analysis**.
    - We revised the section to include more results for different versions of our metric, and we now focus on the 5-way problem, which brings out a lot more differences between methods and is overall more robust.
    - We reworked **Appendix D - Additional Results** to include more discussion about the results across all models, evaluation settings, and metrics.
- **New Appendix B8**. R1 asked about sensitivity to different edits from different authors. CEBaB embeds a subset of examples where the same example and goal were given to two editors, so we can assess this important question. Overall, we find that different edits differ by at most one star in the final ratings.
- **We ran an additional crowdsourcing effort** to validate dataset consistency, verifying that editing for a concept C has negligible to no impact on the ratings for other concepts C’. Details on this study are provided in our response to R1, and we will expand on this study for the next version of the paper.

We believe that the above addresses the major concerns and omissions identified by reviewers, but we are happy to continue engaging with the reviewers, as this process has already helped us make our core findings more robust and accessible.

---

### Meta-Review · Area_Chair_J2PX · 2022-08-25

**Recommendation:** Accept
**Confidence:** Certain

**Metareview:**


The paper presents a new benchmark dataset for assessing explanation methods in NLP, on the sentiment analysis domain. The dataset is unique in that it focuses on the casual effects of modifying specific aspects, providing minimal pairs where only one of the aspects is different. After constructing the benchmark, the paper uses a causality-based metric (Section 2) to evaluate existing explanation methods. The reviewers (NcbX / dBGa) agree that the paper is well motivated and can provide useful resources for the community. The experimental results show a simple baseline they propose performs on par with existing explanation methods, which is already an interesting finding to the community. While the reviewer spotted some flaws (not providing important details, positioning of the work, weak discussion of related work, etc), most seem fixable by camera ready. I’d recommend acceptance.




**Award:**

No

---

### Decision · Program_Chairs · 2022-09-14

Accept